# Effects of tuberculosis and/or HIV-1 infection on COVID-19 presentation and immune response in Africa

Elsa du Bruyn[1,2,19], Cari Stek[1,2,3,19], Remi Daroowala[1,2,3], Qonita Said-Hartley[4], Marvin Hsiao [5,6], Georgia Schafer[1,7,8], Rene T. Goliath[1], Fatima Abrahams[1], Amanda Jackson[1], Sean Wasserman[1,2], Brian W. Allwood[9], Angharad G. Davis[1,10,11], Rachel P.-J. Lai[1,3,10], Anna K. Coussens[1,5,12], Katalin A. Wilkinson[1,2,10,11], Jantina de Vries[2], Nicki Tiffin[1,13,14,15], Maddalena Cerrone[1,3,10], Ntobeko A. B. Ntusi[1,2], HIATUS consortium*, Catherine Riou[1,5,19] ✉ & Robert J. Wilkinson[1,2,3,10,11,19] ✉

Few studies from Africa have described the clinical impact of co-infections on SARS-CoV-2 infection. Here, we investigate the presentation and outcome of SARS-CoV-2 infection in an African setting of high HIV-1 and tuberculosis prevalence by an observational case cohort of SARS-CoV-2 patients. A comparator group of non SARS-CoV-2 participants is included. The study includes 104 adults with SARS-CoV-2 infection of whom 29.8% are HIV-1 co-infected. Two or more co-morbidities are present in 57.7% of participants, including HIV-1 (30%) and active tuberculosis (14%). Amongst patients dually infected by tuberculosis and SARS-CoV-2, clinical features can be typical of either SARS-CoV-2 or tuberculosis: lymphopenia is exacerbated, and some markers of inflammation (D-dimer and ferritin) are further elevated ($p < 0.05$). Amongst HIV-1 co-infected participants those with low CD4 percentage strata exhibit reduced total, but not neutralising, anti-SARS-CoV-2 antibodies. SARS-CoV-2 specific CD8 T cell responses are present in 35.8% participants overall but undetectable in combined HIV-1 and tuberculosis. Death occurred in 30/104 (29%) of all COVID-19 patients and in 6/15 (40%) of patients with coincident SARS-CoV-2 and tuberculosis. This shows that in a high incidence setting, tuberculosis is a common co-morbidity in patients admitted to hospital with COVID-19. The immune response to SARS-CoV-2 is adversely affected by co-existent HIV-1 and tuberculosis.

In 2020, COVID-19 caused by SARS-CoV-2 infection replaced HIV-1 infection and tuberculosis as the leading global infectious cause of death: 4.2 million by 1st August 2021[1]. It is Africa that routinely witnesses the bulk of HIV-1 tuberculosis associated mortality, yet with respect to SARS-CoV-2 the continent has been described as a puzzle[2]. Fewer cases and deaths from COVID-19 than predicted have been notified. This has been attributed to limited molecular testing (serological data suggests infection has been more widespread[3,4]), a much younger population (and thus fewer severe cases and deaths), and the possible influences of pre-existing immunity, genetic factors, and the early implementation of public health measures, especially restriction on already limited intracontinental travel.

A full list of affiliations appears at the end of the paper. *A list of authors and their affiliations appears at the end of the paper. ✉e-mail: cr.riou@uct.ac.za; Robert.Wilkinson@uct.ac.za

In South Africa, importation of SARS-CoV-2 into its Western Cape province in March 2020 led to a substantial first wave peaking in July 2020 that exerted extraordinary demand on hospital services[5, 6]. Over 2.4 million confirmed infections have occurred, and the South African Medical Research Council estimates 326,280 excess deaths nationally since 3rd May 2020[7]. The Western Cape province of South Africa has high HIV-1 prevalence and tuberculosis incidence to which considerable and often co-incident non-communicable disease burden is added with obesity, type 2 diabetes mellitus, hypertension, and vascular comorbidities being very common[8, 9]. Robust provincial health information systems triangulate multiple data sources to enumerate health conditions, with assignment of certainty levels for each enumeration[10]. These systems were deployed rapidly during the COVID-19 first wave to powerfully identify co-morbidities associated with COVID-19 amongst (i) public sector patients, (ii) laboratory-diagnosed COVID-19 cases and (iii) hospitalized COVID-19 cases. Among 3,460,932 patients (16% HIV-1 infected), 22,308 were diagnosed with COVID-19, of whom 625 (2.8%) died. In addition to widely recognised risk factors (male sex, increasing age, diabetes, hypertension and chronic kidney disease) COVID-19 death was associated with HIV-1 (adjusted hazard ratio [aHR] 2.14), with similar risks irrespective of viral load and immunosuppression, and also current and previous tuberculosis (aHR 2.70 and 1.51, respectively)[11]. These analyses were later confirmed in a very large South African national study in which associated were HIV infection (aOR 1.34), past tuberculosis (1.26), and current tuberculosis (1.42)[12].

A number of clinical studies early during the pandemic examined the interaction between HIV-1 and SARS-CoV-2[13–22]. The presentation and outcome of SARS-CoV-2 in HIV-1 co-infected patients were reported as not differing greatly from that in HIV-1 uninfected persons[23–25]. Two reports suggest worse outcomes associated with lower CD4 count[18, 26] and a further report suggests prolonged SARS-CoV-2 shedding (determined by RT-PCR) and reduced anti-SARS-CoV-2 antibody response in those whose antiretroviral therapy (ART) has been interrupted[27]. Amongst patients admitted to hospital in the UK cumulative day-28 mortality from COVID-19 was similar HIV-1 positive versus negative groups (26.7% vs. 32.1), but in those under 60 years of age HIV-positive status was associated with increased mortality (21.3% vs. 9.6%)[28].

For tuberculosis, even less systematic information exists. Motta et al. reported coincident tuberculosis in their own, and a prior, cohort of, co-infections concluding tuberculosis might not be a major determinant of mortality[29, 30]. Stochino et al. reported a case series of what appears to be nosocomial transmission of SARS-CoV-2 amongst tuberculosis inpatients noting tuberculosis outcomes to be generally unaffected[31]. In a study of 159 South African children with SARS-CoV-2, 51 of 62 hospitalised were symptomatic and of these 7/51 had a recent or current diagnosis of tuberculosis[32]. In a meta-analysis Sarkar and colleagues concluded patients with tuberculosis have an increased risk of mortality during co-infection with SARS-CoV-2 (risk ratio = 2.10; 95% CI, 1.75–2.51)[33]. A postmortem surveillance study conducted in Zambia reported the presence of tuberculosis in 22/71 (31%) patients who died with a positive RT-PCR for SARS-CoV-2[34].

We therefore conducted a facility-based observational study to investigate the interaction and overlap between SARS-CoV-2, HIV-1 and *M. tuberculosis* infections. The purpose was to describe the clinical presentation, radiographic appearances, clinical laboratory features, and the immune response of patients hospitalized with proven or suspected SARS-CoV-2 infection in an African setting not only endemic for HIV-1 and tuberculosis but in which other non-communicable co-morbidities are common[9].

## Results
### Participants included in the analysis
One hundred and four SARS-CoV-2 RT-PCR positive, and 50 negative, participants (non-COVID-19 other disease controls, NC) were enrolled.

Eight of the 50 RT-PCR negative NC participants tested SARS-CoV-2 antibody positive and were thus excluded from further analysis. The baseline characteristics of the remaining 146 participants are shown in Table 1, with the final diagnosis of NC participants being shown in Supplementary Table 1. COVID-19 cases and NC were similar in age, sex, percentage HIV-1 or tuberculosis co-infected (29.8% and 14.4% versus 30.9% and 11.9%, respectively) and days of illness prior to blood sampling. HIV-1 co-infected NC had a lower CD4 count (median: 18 cells/mm³

**Table 1 | Clinical characteristics of COVID-19 versus hospitalized non-COVID-19 participants**

| | COVID-19 (n = 104) | non-COVID-19 NC (n = 42) |
|---|---|---|
| Age (median, IQR) | 53 [44–61] | 51 [36–66] |
| Male (%) | 52.9% (n = 55) | 35.7% (n = 15) |
| HIV-1 co-infected (%, n) | 29.8% (n = 31) | 30.9% (n = 13) |
| On antiretroviral therapy (ART) | 74.2% (n = 23) | 46.1% (n = 6) |
| Time on ART (years) ᵃ | 9.6 [6–12] | 3.6 [0.4–11] |
| CD4 count (cells/mm³) ᵃ | 132 [51–315] | 18 [7–102] |
| CD4 count (cells/mm³) in ART established | 188.5 [76–336] | 137 [5–780] |
| CD4 count (cells/mm³) in ART naïve | 66 [14–106] | 18 [7–55] |
| Log₁₀ HIV Viral load ᵃ | <1.3 [<1.3–4.14] | 5.36 [2.49–5.52] |
| Log₁₀ HIV viral load in ART established | <1.3 [<1.3–<1.3] | 3.08 [<1.3–5.4] |
| Log₁₀ HIV viral load in ART naïve | 4.44 [4.13–4.8] | 5.44 [5.0–5.56] |
| *M. tuberculosis* positive (%, n) | 14.4 % (n = 15) | 11.9% (n = 5) |
| Previous tuberculosis episode/s (within 5 years) | 6.7% (n = 7) | 9.5% (n = 4) |
| Co-morbidities (%, n) | | |
| Cardiovascular | 6.7% (n = 7) | 42.9% (n = 18) |
| Hypertension | 48.1% (n = 50) | 54.8% (n = 23) |
| Diabetes | 39.4% (n = 41) | 28.6% (n = 12) |
| Obesity | 32.7% (n = 34) | 33.3% (n = 14) |
| Other respiratory diseases | 6.7% (n = 7) | 26.2% (n = 11) |
| SARS-CoV-2 PCR positive | 100% | 0% |
| SARS-CoV-2 serology positive ᵇ | 69.2% (n = 72) | 0% |
| Cut-off index (median, IQR) ᶜ | 7.06 [0.38–25.78] | 0.07 [0.07–0.08] |
| WHO COVID-19 ordinal scale at enrolment (%, n) | | |
| 2* | – | 16.7% (n = 7) |
| 3 | 17.3% (n = 18) | 42.8% (n = 18) |
| 4 | 36.5% (n = 38) | 40.5% (n = 17) |
| 5 | 28.8% (n = 30) | – |
| 6 | 16.3% (n = 17) | – |
| 7 | 0.96% (n = 1) | – |
| Mild and moderate (WHO < 5, %) | 53.8% (n = 56) | 100% |
| Severe (WHO ≥ 5, %) | 46.2% (n = 48) | 0% |
| On steroid treatment (%, n) | 78.8% (n = 82) | 26.1% (n = 11) |
| Thrombo-embolic complications (%, n) | 9.6% (n = 10) | 0% |
| Days with symptoms at sampling ᵃ | 9 [6–14] | 8 [3–18] |
| Overall days in clinical care ᵃ | 13 [7–24] | 7 [4–13] |

NC: Non-COVID-19 participants, IQR: Interquartile range.
*Seven patients who were not hosptialised following outpatient assessment.
ᵃMedian and [IQR] ᵇ Two values were missing ᶜ SARS-CoV-2 serology was performed using the Roche Elecsys assay, measuring SARS-CoV-2 nucleocapsid-specific antibodies. Results are reported as numeric values in form of a cut-off index (signal sample/cut-off), where a COI < 1.0 corresponds to non-reactive plasma and COI ≥ 1.0 to reactive plasma.

[IQR: 7–102] versus 132 [51–315], $p = 0.028$) and higher median HIV-1 viral load (median: 5.36 $\log_{10}$ HIV RNA copies/ml [IQR: 2.49–5.21] versus <1.3 [<1.3–4.14], $p = 0.0005$). Although the overall frequency of co-morbidities was similar between COVID-19 patients and NC (Supplementary Fig. 1), NC were more likely to exhibit cardiovascular or other respiratory comorbidities (6.7% and 6.7% versus 42.9 and 26.2%, respectively). The majority of participants in both groups had two or more co-morbidities (COVID-19 57.7% and NC 74.4%). Forty six % COVID-19 patients were assessed as grade 5 or above on the WHO ordinal scale for COVID-19. There was greater use of adjunctive corticosteroid therapy (78.8% versus 26.1%, $p < 0.0001$) and a longer hospital stay (13 versus 7 days, $p = 0.0008$) in COVID-19 patients.

### Mortality due to COVID-19

The analysis was neither designed nor powered to determine risk factors for death amongst COVID-19 patients. 30 deaths occurred (29% of the total diagnosed SARS-CoV-2 RT-PCR positive) of which 23 (77%) were males, the median age was 55 years (IQR, 46–66), 25 (83%) were classified as WHO grade 5 or above, and 28 (93%) received corticosteroid therapy (Supplementary Table 2). Twenty % patients who died were HIV-1 co-infected, 20% had tuberculosis and 10% had the triple combination of SARS-CoV-2, tuberculosis and HIV-1. 6/15 (40%) of patients with coincident tuberculosis died of whom equal numbers were HIV-1 infected and uninfected (Supplementary Tables 2 and 3).

### Radiographic features

Chest radiographs at enrolment were evaluated by three scoring systems: Brixia[35], British Society for Thoracic Imaging (BSTI)[36] and the percentage of unaffected lung. There was a close relationship between Brixia and BSTI radiographic severity scores (Supplementary Fig. 2A); and an inverse relationship between both the BSTI and Brixia scores and the percentage of unaffected lung (Supplementary Fig. 2B, C). The percentage of unaffected lung was lower in COVID-19 patients than NC (Median 35 vs 70%, $p < 0.0001$, Supplementary Fig. 2D). There was no significant effect of either HIV-1 or tuberculosis co-infection alone on the percentage unaffected lung in COVID-19 patients, but patients with triple infection tended to exhibit a higher percentage of unaffected lung ($p = 0.007$, Supplementary Fig. 2E).

### Clinical features of COVID-19 in participants with co-incident tuberculosis and/or HIV-1 infection

A complete listing of patients with both tuberculosis and COVID-19 (+/- HIV-1 infection) is provided in Supplementary Table 3 and examples of radiographic appearances are provided in Fig. 1A, B and Supplementary Fig. 3A–D. In COVID-19 patients, both Brixia and BSTI radiographic scores related closely to the WHO clinical severity score (Fig. 1D, E). There was a trend towards a greater proportion of radiographs classified as non-COVID-19 like in those with COVID-19 and coincident tuberculosis, regardless of HIV-1 status (Fig. 1C). In HIV-1-co-infected, tuberculosis-positive, SARS-CoV-2 positive patients the Brixia score was lower than other patient groups. This was significant for both singly SARS-CoV-2 positive patients and HIV-1 uninfected, tuberculosis-positive, SARS-CoV-2 positive patients (median score: 8 vs. 14 and 15.5, $p = 0.006$ and 0.038, respectively, Fig. 1F). These data therefore indicate that in persons SARS-CoV-2 RT-PCR positive, radiographic appearances were most often typical of COVID-19 with the finding of tuberculosis being incidental, but that in a smaller number of cases appearances were predominantly of tuberculosis, with SARS-CoV-2 being co-prevalent. HIV-1 is known to be associated with non-specific radiographic appearances of tuberculosis[37] and appears in triple combination to associate with atypical appearances also of COVID-19 that were ostensibly less severe. This impression was supported by analysis of WHO ordinal scale at presentation, which was slightly but significantly lower than that of HIV-1 uninfected SARS-CoV-2 monoinfected patients (median 5, IQR 4–6) in the presence of HIV-1

alone (median 4, IQR 43–5, $p = 0.038$) and in HIV-1 and tuberculosis co-infected SARS-CoV-2 participants (median 4, IQR 3-4, $p = 0.008$).

### Peripheral white cell count

Peripheral blood data stratified by COVID-19 status and WHO severity (3–7) are shown in Fig. 2. Data from 72 HIV-1 uninfected healthy controls (HC) recruited to pre-pandemic studies (pre-2019) were also used for comparison (Supplementary table 4). The total white cell count was raised in both NC and COVID-19 patients by comparison with HC with no differences between the patient groups, and the count increased with increasing COVID-19 severity (Fig. 2A). Amongst COVID-19 patients, the total white cell count was highest in HIV-1 uninfected patients with coincident tuberculosis (median: 15.34 × 10$^9$ /L, IQR: 11.97–17.13). Lymphocyte counts were lower in both NC and COVID-19 patients (with no differences between these groups) compared to HC. The count trended towards a decrease with increasing COVID-19 severity and was significantly lowest in triply infected patients ($p = 0.03$, Fig. 2B). Both neutrophil and monocyte counts were raised in both NC and COVID-19 patients with no differences between these groups. Neutrophil, but not monocyte, count related to increasing COVID-19 severity (Fig. 2C, D). Those with coincident tuberculosis had higher monocyte counts than those with combined HIV-1 and COVID-19 infections without tuberculosis ($p = 0.04$, Fig. 2D). By comparison with HC, eosinophil counts were lower in both NC and COVID-19 patients were lowest in those with COVID-19 infection with a trend observed in relation to disease severity ($p = 0.004$, Fig. 2E).

### Further analysis of lymphopenia by flow cytometric analysis

We further assessed the nature of lymphopenia by flow cytometric analysis of whole blood. There was strong correlation between absolute CD4 numbers and the % CD4 determined by flow cytometry (Supplementary Fig. 4A). Amongst COVID-19 patients increasing disease severity correlated with a decrease in CD4 and CD8 positive lymphocytes (Supplementary Fig. 4B). To investigate lymphocyte differences in COVID-19 patients, additional values were obtained from a subset of 118/163 similar ambulant participants enrolled to a prior study of HIV-1 uninfected and infected persons with either immune evidence of tuberculosis sensitization but no symptoms or microbiologically confirmed pulmonary tuberculosis treated at a community health clinic (Supplementary Table 4 and[38, 39]). When compared to HIV-1 uninfected healthy persons, the percentage of lymphocytes positive for CD4 was lower in HIV-1 uninfected NC ($p \leq 0.03$) and more so in COVID-19 patients ($p < 0.0001$, Supplementary Fig. 4C). Amongst HIV-1 infected patients the values were especially low for NC. Amongst patients with coincident HIV-1 and tuberculosis infection, % of CD4 positive lymphocytes was also very low irrespective of the presence or absence of SARS-CoV-2 infection. When compared to HIV-1 uninfected HC, the percentage of lymphocytes positive for CD8 was also lower in both HIV-1 uninfected NC and more so in COVID-19 patients (Supplementary Fig. 4C). However, this pattern was not observed amongst HIV-1 infected patients. The % CD8 positive lymphocytes was markedly depressed in HIV-1 uninfected patients with coincident tuberculosis and SARS-CoV-2 infection (median: 11.2%, IQR: 10.2–15.1). Amongst HIV-1 uninfected patients with COVID-19 there was a significant trend towards a lower percentage of lymphocytes positive for CD4 and CD8 (Supplementary Fig. 4B) with increasing WHO grade severity.

### Serum biomarkers

A number of soluble biomarkers have been associated with COVID-19 and its severity[40]. In our study, higher C-reactive protein, D-dimer, lactate dehydrogenase (LDH) and ferritin levels were observed in COVID-19 patients compared to NC (Fig. 3A–D). In the case of ferritin, this difference approached 10-fold higher (1116 µg/L (IQR: 643–1974) vs 127 µg/L (IQR: 77–477), respectively, $p < 0.0001$. However, apart from LDH (Fig. 3D), there was no significant trend in any of these

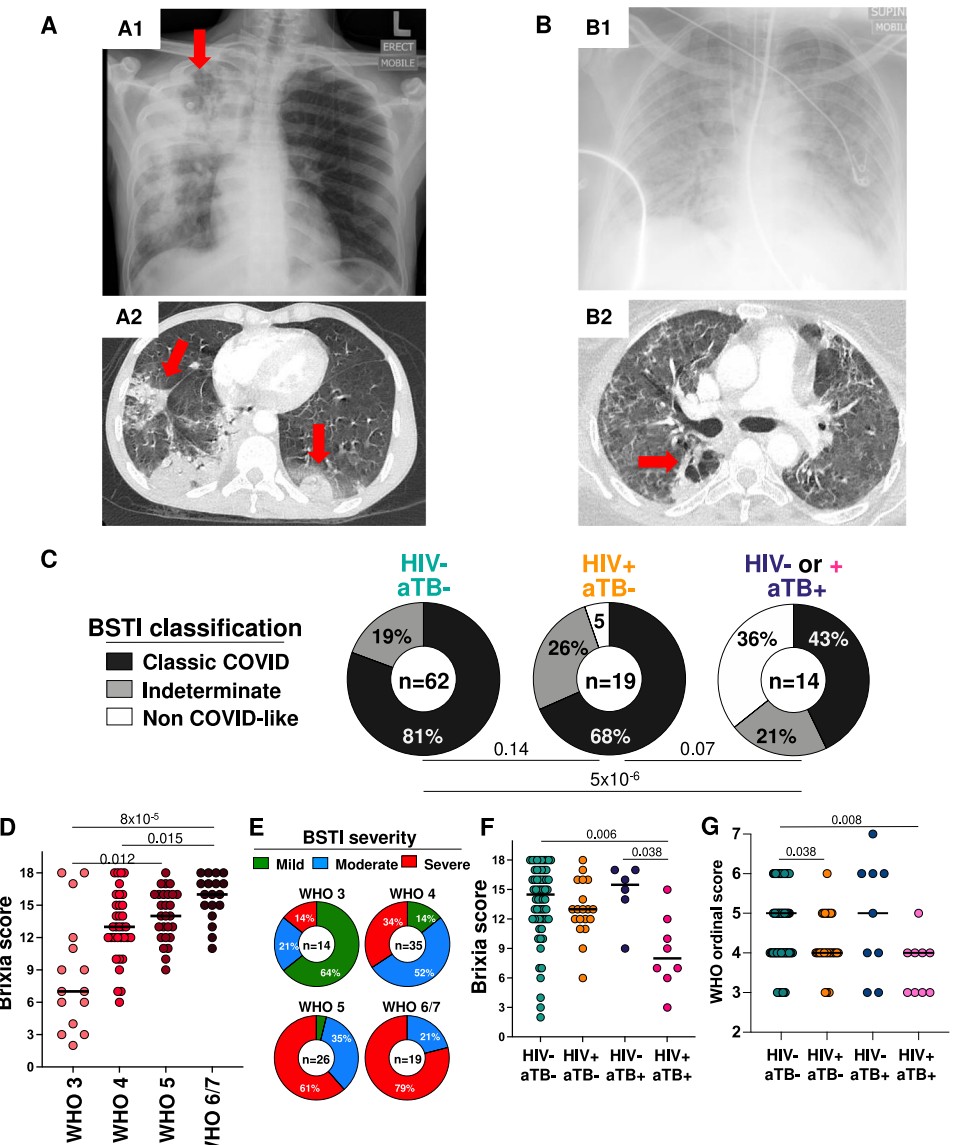

**Fig. 1 | Radiographic appearances of combined SARS-CoV-2 and *M. tuberculosis* infection and relationship between radiographic (Brixia and British Society for Thoracic Imaging (BSTI)) score and clinical severity assessed by WHO COVID-19 ordinal scale. A, B** Chest radiographs and computed tomographic pulmonary angiography (CTPA) of two COVID-19 patients. A1: 43-year-old HIV-1 uninfected male (patient number 58) presenting with consolidation and cavitation in the right upper lobe (red arrow), 3+ sputum smear positive and also SARS-CoV-2 RT-PCR positive (threshold cycle: 32.45). A2: Because of persistent hypoxia and tachypnoea he underwent CTPA which showed bi-basal wedge-shaped opacities in keeping with pulmonary embolism (PE) and/or consolidation related to COVID-19. Discharged after 22 days. B1: 43-year-old HIV-1 uninfected female (patient number 36) presenting with SARS-CoV-2 positive RT-PCR (threshold cycle: 21.2) with diffuse bilateral pulmonary opacification with ground glass and consolidation on the chest radiograph who deteriorated necessitating intubation and ventilation for 36 days. The patient remained O2 dependent after extubation and a CTPA (B2) for suspected PE instead revealed cavitation associated with opacification and air

bronchograms in the superior segment of the right lower lobe (arrow) together with subcarinal and pretracheal lymphadenopathy. The patient was found Gene Xpert MTB/Rif positive. **C** BSTI classification of RT-PCR proven SARS-CoV-2 cases in the absence or presence of HIV-1 and/or tuberculosis. The increase in the proportion of radiographs classified as Non-COVID like tended to increase in those with coincident tuberculosis. Comparisons were performed by Chi-square test. **D** Brixia radiographic and, **E** BSTI, severity scores in relation to WHO clinical severity scale in $N = 104$ SARS-CoV-2 participants. Comparisons were performed using a Kruskal-Wallis test. **F** Brixia score in relation to the presence or absence of HIV-1 and/or tuberculosis co-infection in N = 104 SARS-CoV-2 participants. The extent of changes related COVID-19 was decreased amongst those with coincident HIV-1 associated tuberculosis. Comparisons were performed using a Kruskal-Wallis test with Dunn's correction. **G** WHO ordinal scale at presentation in $N = 104$ SARS-CoV-2 participants in relation to the presence or absence of HIV-1 and/or tuberculosis co-infection. Comparisons were performed using a Kruskal-Wallis test with Dunn's correction.

markers in relation to COVID-19 severity. HIV-1-infected patients with both COVID-19 and tuberculosis had higher levels of D-dimer and ferritin than those HIV-1 infected with COVID-19 alone. Amongst COVID-19 patients there was strong positive correlation between total white cell (WCC) and neutrophil counts (rho = 0.95, $p < 0.0001$); moderate correlation between ferritin and LDH, and between lymphocyte and monocyte counts (Fig. 3E). The relationship between

(WCC) and neutrophil counts, and between ferritin and LDH was also observed in NC (Fig. 3F).

## SARS-CoV-2 specific antibody response
We next determined relationships between disease severity amongst COVID-19 patients, the presence of HIV-1 and/or tuberculosis as comorbidities, and the level of anti-SARS-CoV-2 nucleocapsid-specific

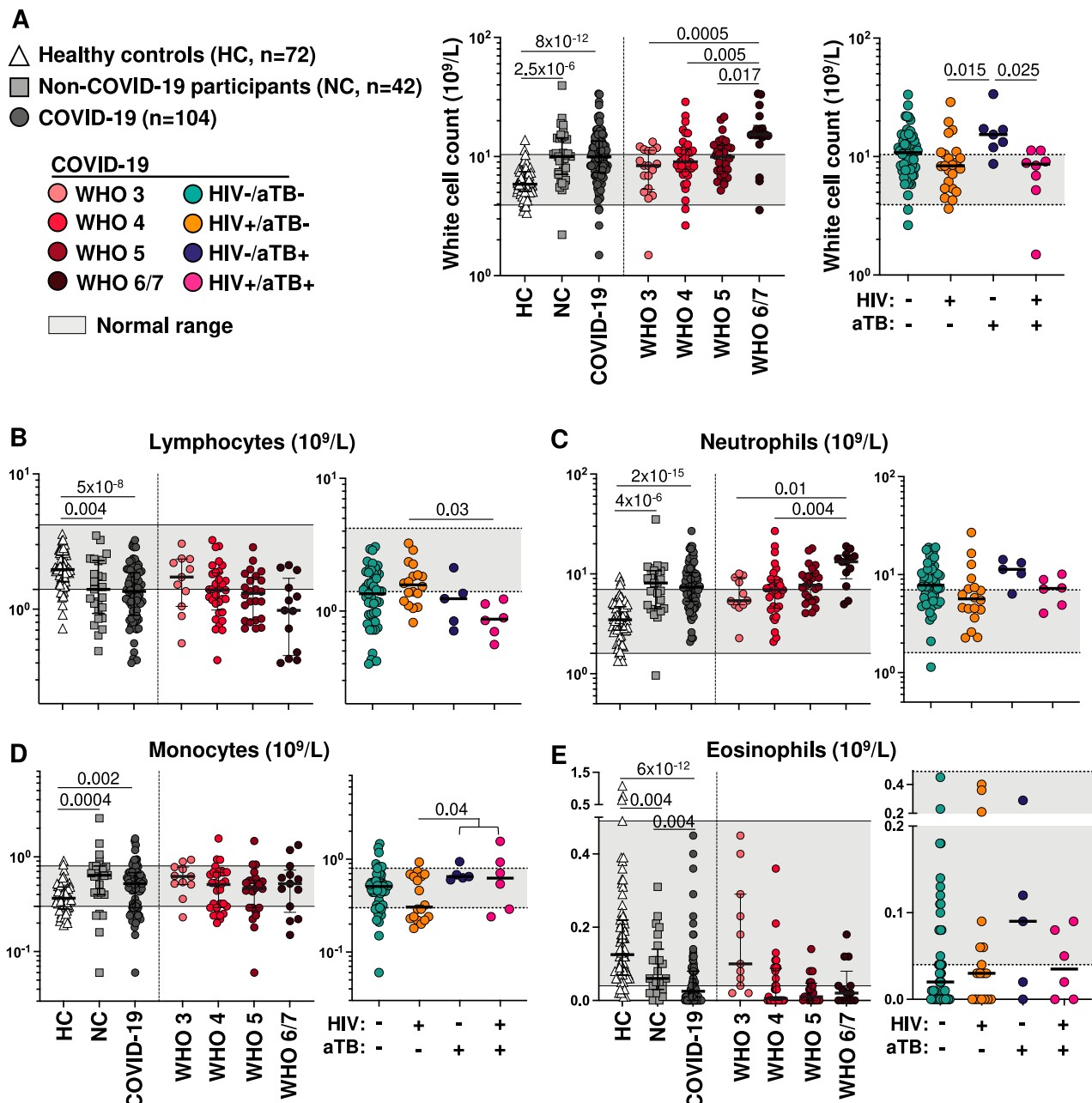

**Fig. 2 | Total white cell and differential count in healthy persons (HC), non-COVID-19 hospitalized participants (NC), and COVID-19 participants. A** The total white cell count was raised in both COVID-19 and NC patients with no differences between the groups. The count increased with increasing COVID-19 severity. Amongst COVID-19 patients the total white cell count was highest in HIV-1 uninfected patients with coincident tuberculosis. **B** The lymphocyte count was lower in both COVID-19 and NC patients with no differences between these groups. The count tended to decrease with increasing COVID-19 severity. Amongst COVID-19 patients the lymphocyte count was lowest in triply infected patients with HIV-1 and coincident tuberculosis. **C** The neutrophil count was raised in both NC and COVID-19 patients with no differences between the groups. The count increased with increasing COVID-19 severity. Amongst COVID-19 patients there were no

differences in count in the presence of HIV-1 and/or tuberculosis. **D** The monocyte count was raised in both NC and COVID-19 patients with no differences between these groups. No trend in relation to COVID-19 severity was observed. Amongst COVID-19 patients, those with coincident tuberculosis had higher counts than those with combined HIV-1 and COVID-19 infections without tuberculosis. **E** Eosinophil counts were lower in both COVID-19 and NC patients being lowest in those with COVID-19 infection with a trend in this decrease observed in relation to disease severity. Amongst COVID-19 patients, no significant effect of HIV-1 and/or tuberculosis infection was observed. Line indicates the median value and shaded area the normal range. All comparisons were performed using a Kruskal-Wallis test with Dunn's correction.

antibody levels determined by the Roche Elecsys assay[41]. We have previously reported increased antinucleocapsid antibody levels in COVID-19 patients with more severe disease although no overall association with survival was noted[42]. Although no significant quantitative difference in cut-off index was observed by HIV-1 and or tuberculosis status, 8/15 tuberculosis patients in total and 5/8 with HIV-1

and tuberculosis were below the threshold for positivity (Fig. 4A). There was a significant trend towards decreased antibody levels in participants with a lower percentage CD4 lymphocytes with just 3/10 antibody responders amongst patients with a CD4 percentage of lymphocytes below 10% (Fig. 4B). We have previously reported positive correlation between the percentage of CD4 T cells and the

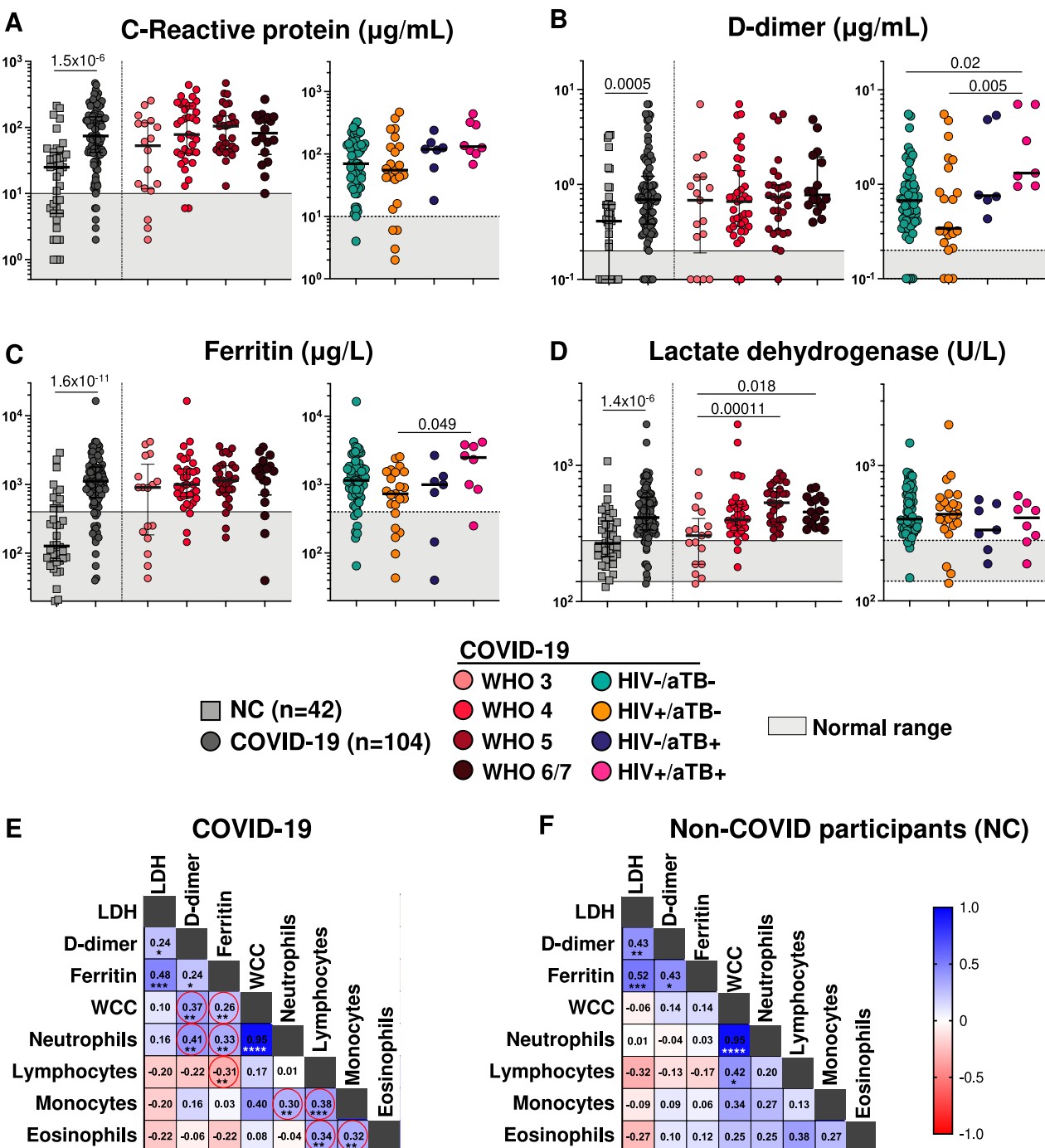

**Fig. 3 | Serum biomarkers in relation to disease status and severity, and correlation between those markers and peripheral cell counts. A–C** C-Reactive protein (CRP), D-dimer and ferritin levels showed similar trends being significantly higher in COVID-19 patients than NC. There was no significant trend in relation to severity of COVID-19 disease. HIV-1 infected patients with both COVID-19 and tuberculosis had higher levels of all three markers than those with COVID-19 alone. Comparisons between the NC and COVID-19 groups were performed using a Mann Whitney test and other comparisons were performed using a Kruskal-Wallis test. **D** Lactate dehydrogenase was significantly higher in COVID-19 patients than NC, with levels tending to increase with increasing COVID-19 disease severity. There was however no relationship with HIV-1 and/or tuberculosis status. Comparisons

between the NC and COVID-19 groups were performed using a Mann Whitney test and other comparisons were performed using a Kruskal-Wallis test (two sided). **E, F** Correlation matrix of serum biomarkers in COVID-19 patients (E) and NC participants (F). Amongst COVID-19 patients there was strong positive correlation between total white cell (WCC) and neutrophil counts, and moderate between ferritin and LDH, and between lymphocyte and monocyte counts. In NC participants, the relationship between total white cell (WCC) and neutrophil counts, and between ferritin and LDH was also observed. Positive correlations are indicated in blue and negative correlation in red. Spearman R-values are indicated in each square. *$P < 0.05$, **$P < 0.01$, ***$P < 0.001$, ****$P < 0.0001$. Line indicates the median value and shaded area the normal range.

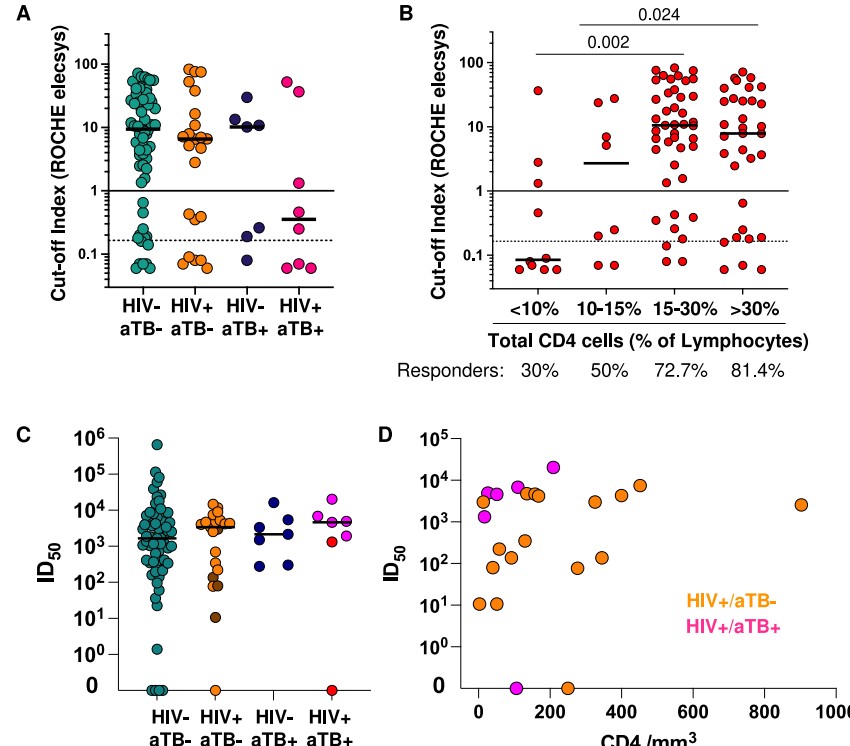

**Fig. 4 | Serum anti-nucleocapsid IgG levels in $n = 104$ COVID-19 patients in relation to the presence of HIV-1 and/or tuberculosis co-infection. A** Although no significant quantitative difference in cut-off index was observed by HIV-1 and or tuberculosis status, 8/15 tuberculosis patients in total and 5/8 with HIV-1 and tuberculosis were below threshold for positivity of 1.0. The >1 cut-off value represents the manufacturer's cut-off. The >0.165 cut-off value represents the optimal cut-off value defined by ROC curve analyses on 197 participants that improved the performance of the test to give a sensitivity of 100% (95%CI: 94.0-100%)[57]. **B** There was a significant trend towards decreased antibody levels in participants with Lower percentage CD4 lymphocytes. Comparisons were performed using a Kruskal-Wallis test. **C** No significant difference in ability to neutralize SARS-CoV-2 pseudovirus between groups was observed although a minority not prescribed antiretroviral therapy (darker shading) tended to have lower $ID_{50}$ values. **D** In subset of $n = 24$ HIV-1 co-infected patients for whom the absolute CD4 count was available, there was no trend between neutralising antibody level and CD4 ($p = 0.34$) and patients with tuberculosis (pink) did not form a distinct subgroup. Correlation was tested by a two-tailed non-parametric Spearman rank test.

antibody cut-off index with triply infected patients tending to show low levels of both parameters[42]. We therefore further investigated in a subset of 91 SARS-COV-2 positive (64 HIV-TB-, 27 HIV + TB-, 7 HIV-TB + , 7 HIV + TB + ) for whom adequate sample remained available the serum neutralization of spike-containing SARS-CoV-2 pseudovirus. No significant difference in ability to neutralize SARS-CoV-2 pseudovirus between groups was observed although a minority not prescribed antiretroviral therapy (darker shading) tended to have lower $ID_{50}$ values Fig. 4C). In subset of HIV-1 co-infected patients for whom the absolute CD4 count was available, the trend between neutralising antibody level and CD4 was not significant (r = 0.21 $p = 0.34$) and patients with tuberculosis (pink) did not form a distinct subgroup (Fig. 4D).

### SARS-CoV-2 specific CD8⁺ T cell response

We have previously reported the frequency and phenotype of SARS-CoV-2 antigen-specific CD4 T cells in acute COVID-19 infection[42]. In experimental models and in humans SARS-CoV-2 specific CD8 T cells are also thought to contribute to protection[43, 44]. In 95/104 patients we therefore assayed the SARS-CoV-2 specific CD8 response to peptides spanning the M, N and S sequences and compared it to the CD4 response (Fig. 5A, B and Supplementary Fig. 5A, B). 35.8% participants had a detectable CD8 response which was invariably accompanied by a CD4 response, which was found in 83.2% participants. Although CD8 responses were thus significantly less often detectable ($p < 0.001$), their magnitude when present was similar to that of the CD4 response (Fig. 5B). The frequency of CD8 responses did not relate to disease severity of outcome (Supplementary Fig. 5C) but these cells were undetectable in participants with coincident HIV-1 and tuberculosis (Fig. 5C). The frequency of polyfunctional triply positive Interferon-gamma, IL-2 and TNF-alpha cytokine producing cells was associated with milder, and singly positive interferon-gamma positive with more severe, SARS-CoV-2 (Fig. 5D). We also compared the differentiation status of SARS-CoV-2 responding CD4 and CD8 cells (gating strategy shown in Supplementary Fig. 5D) and found evidence of terminal effector differentiation in CD8 cells which were more frequently positive for HLA-DR, CD38, Granzyme B and PD-1 than CD4 cells (Fig. 5E). There was correlation between the expression of HLA-DR, Ki-67 and CD38 on antigen specific CD4 and CD8 subsets (Supplementary Fig. 5E).

### Discussion

We have provided one of the first accounts of the clinical presentation and laboratory features of COVID-19 in an African setting in which there is high non-communicable, but also communicable (HIV-1 and tuberculosis) co-morbidity. An important conclusion is that the triple occurrence of SARS-CoV-2, HIV-1 and tuberculosis is not uncommon. Clinical features may be dominated by either SARS-CoV-2 or tuberculosis and, although overall radiographic severity may be modulated, lymphopenia is exacerbated, and some markers of inflammation are even more elevated in such patients. Forty percent of patients with coincident COVID-19 and tuberculosis died.

Our study benefitted inclusion of participants admitted to the same hospital at the same time as the surge in COVID-19 admissions

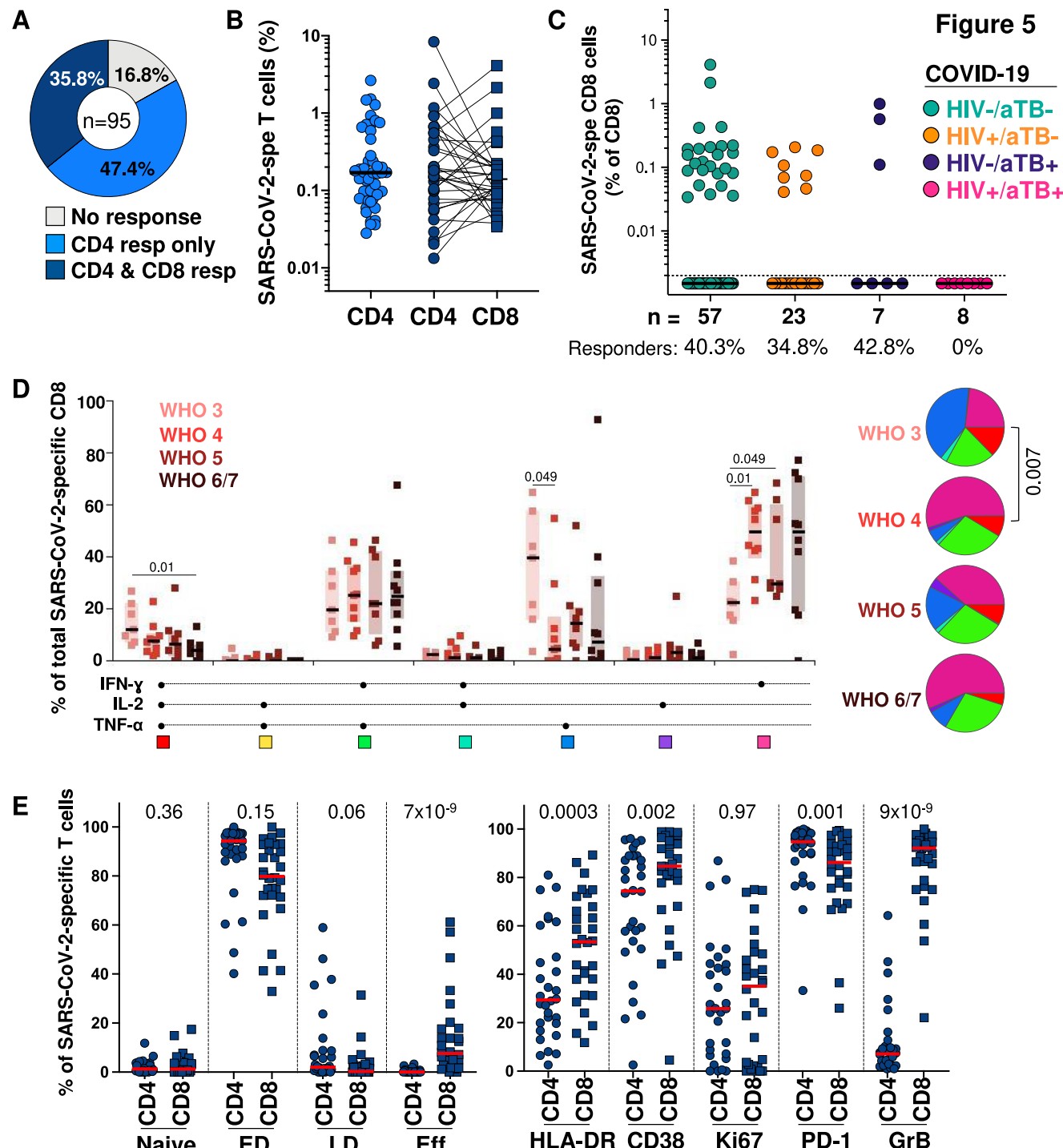

**Fig. 5 | SARS-CoV-2 antigen-specific CD8 T cell response in COVID-19 patients.**
**A** Proportion of COVID-19 patients (*n* = 95) exhibiting a detectable SARS-CoV-2 CD8 T cell response. **B** Comparison of the magnitude of SARS-CoV-2-specific CD4 and CD8 T cells in COVID-19 patients exhibiting a SARS-CoV-2 CD4 T cells response in in absence of CD8 responses (*n* = 45, light blue) and in patients exhibiting both a CD4 and CD8 SARS-CoV-2 response to SARS-CoV-2 (*n* = 34, dark blue). **C** Prevalence and frequencies of SARS-CoV-2-specific CD8 T cells in 95 COVID-19 patients stratified by HIV and/or tuberculosis co-infection. **D** Polyfunctional profile of SARS-CoV-2-specific CD8 T cells in 34 COVID-19 patients, stratified by WHO score and outcome. Line indicates the median value and shaded area the normal range. A two-sided Wilcoxon rank test was used to compare response patterns between groups. **E** Comparison of the memory differentiation profile (left) and activation profile (right) between SARS-CoV-2-specific CD4 T cells and SARS-CoV-2-specific CD8 T cells. *N* = 95 patients, statistical comparisons were calculated using a two-sided Mann-Whitney test.

who were subsequently assigned alternative diagnoses with negative virological and serological tests for SARS-CoV-2. A limitation is that this is not however a formal control group (or reported as such) and we cannot exclude, under the circumstances, that such tests were false negative. In addition, their ordinal score (which is dominated by consideration of the need for, and type of, respiratory support) was lower and it could be argued differences in biomarkers may related to severity and may not be specific to SARS-CoV-2 infection. However, the

existence of this group allowed us to compare blood markers that have been used to characterise and prognosticate SARS-CoV-2 infection. We found that neither raised total white count, neutrophilia or lymphopenia can be inferred as specific for COVID-19 (Fig. 2). By contrast, the plasma levels of several markers, including the CRP, D-dimer, LDH and especially ferritin were markedly elevated in COVID-19 patients (Fig. 3). Of these markers however, only LDH showed an increasing trend with COVID-19 disease severity. A potential confounder of results on peripheral blood markers is the greater use of corticosteroid therapy at the time of sampling in COVID-19 patients. The results of the Recovery trial were announced shortly after we began recruitment to our study and were accompanied by an overnight change in clinical guidance to prescribe such therapy to those severely ill[45]. Notwithstanding this potential blunting of values, our results again reflect the marked severe systemic effects of acute SARS-CoV-2 infection that are very widely documented. A greater proportion of participants who died also received steroids. This therapy is indicated in severe COVID-19[45] and this association is therefore probably confounded by the fact that more severely ill participants were more likely to receive this treatment.

Our analysis tends towards the conclusions of others that have included HIV-1 infected persons in not documenting major differences in presentation and outcome except in those profoundly immunosuppressed[46]. ART use was variable and not equally distributed between COVID-19 cases and NC with the latter more likely to be ART-naïve: perhaps reflecting late presentations of HIV-1 infection at a time when pressure on health services was acute. This feature also precludes meaningful and well-powered comparisons according to ART status. Overall, HIV-1 infected patients recruited to this study appeared not more likely to die. There was a predictable decrease in CD4 T cell numbers and percentage in HIV-1 infected participants, not reflected however in CD8 counts. There were relationships between the peripheral CD4 percentage (and a trend for absolute CD4) and acute antibody levels. Whilst this may reflect the duration of COVID-19 illness and its severity (both tending to increase antibody levels), it remains a concern that both natural and vaccine-induced immune responses to SARS-CoV-2 may be less effective in persons immunosuppressed by HIV-1, especially in those not prescribed ART[47–49].

Our study confirms that a wide range of now well-recognised comorbidities (most commonly hypertension, type 2 diabetes mellitus and obesity) that associate with COVID-19 severity and death were present in the bulk of our cases. Overall, 29% COVID-19 patients died. SARS-CoV-2 acutely brings into focus the interaction between infectious and non-communicable conditions in Africa and the necessity that multimorbidity research must encompass such interaction much more in the future.

Another study limitation is the analysis was not designed or powered to reproduce the previously documented relationships between HIV-1 co-infection and tuberculosis and risk of death from COVID-19[11]. The intent was rather to describe clinical presentation. Against an overall study mortality from COVID-19 of 28.8%, 19.4% of those HIV-1 co-infected, 40% of those with coincident tuberculosis, and 3/8 (37.5%) with coincident HIV-tuberculosis, died. Lymphocyte numbers were lowest, some inflammatory markers were higher and anti-SARS-CoV-2 antibody responses were most depressed, and SARS-CoV-2 antigen-specific CD8 responses undetectable, in those exhibiting triple infection. The late detection of unsuspected tuberculosis in several otherwise typical COVID-19 patients is a cause for concern and increased clinical vigilance. We have documented here and elsewhere that tuberculosis may adversely impact both antibody and T-cell responses to SARS-CoV-2[42]. Additionally, the immunological milieu set up in the lung by SARS-CoV-2, to which corticosteroid therapy will add, may have contributed to reactivation of subclinical or latent tuberculosis[50]. Thus, in environments in which tuberculosis is common, clinical suspicion of presence and also that SARS-CoV-2 may also co-exist in typical presentations of tuberculosis are the most important clinical lessons from our work.

## Methods

### Participant enrolment

This was a single-centre observational case-control study of adult (age >18 years) patients admitted to Groote Schuur Hospital, Cape Town, South Africa with RT-PCR proven, or suspected, SARS-CoV-2 infection. Between 11th June and 28th August 2020, the study team assessed suspected or confirmed SARS-CoV-2 (by nasopharyngeal or oropharyngeal aspirate) participants and expedited written electronic consent for inclusion was provided. Where participants did not have capacity to provide informed consent, e.g. in the case of those admitted in the intensive care unit, informed consent was sought from family members. Informed consent was provided by all participants who regained their capability to provide informed consent after incapacity. Known HIV-1 status or willingness to undergo testing was an inclusion criterion. During the initial surge of hospital admissions recruitment could not be sequential due to caseload. Emphasis was on known SARS-CoV-2 PCR positive inpatients in a variety of dependency settings ranging through person under investigation (PUI), low dependency known positive, high-flow nasal oxygen and intensive care settings. As caseload decreased emphasis on PUI recruitment increased such that appropriate inpatient controls without laboratory evidence of SARS-CoV-2 infection (Non-COVID-19 participants, NC) including seven who were not hospitalised could be recruited. The principal reason to include the NC was to evaluate the potential specificity of white cell counts and inflammatory markers in a contemporaneous group of patients. Data were entered directly into an electronic REDCap data entry system, hosted by the University of Cape Town[51]. The initial aim was to recruit equal numbers of HIV-1-co-infected and -uninfected participants. Participants presenting as persons under investigation (PUI) for SARS-CoV-2 but persistently negative by RT-PCR (Seegene, Roche or Gene Xpert) for SARS-CoV-2 were subsequently serologically tested. In the absence of either RT-PCR or serological evidence of SARS-CoV-2 infection and in the presence of an alternative diagnosis such participants were designated controls. The WHO ordinal severity score was used to categorize participants on admission according to the following categories:

1. Not hospitalized, no limitation on activities;
2. Not hospitalized, limitation on activities;
3. Hospitalized, not requiring supplemental oxygen;
4. Hospitalized, requiring supplemental oxygen by mask or nasal cannulae;
5. Hospitalized, on non-invasive ventilation or high flow oxygen devices;
6. Hospitalized, on invasive mechanical ventilation;
7. Hospitalized, on invasive mechanical ventilation with other organ support.
8. Death.

The study was conducted according to the declaration of Helsinki, conformed to South African Good Clinical Practice guidelines, and was approved by the University of Cape Town's Health Sciences Research Ethical Committee (HREC 207/2020).

For some analyses we included data from HIV-1 uninfected and infected persons with either immune evidence of tuberculosis sensitization but no symptoms (latent tuberculosis, LTBI) or microbiologically confirmed pulmonary tuberculosis who had been recruited to prior studies (HREC 050/2015)[38, 39]. The details of these persons are presented in Supplementary Table 4.

### Sample size

The intent was to include 120 confirmed COVID-19 patients (equal numbers with or without HIV-1 co-infection) and 40 COVID-19 negative

participants (equal numbers with or without HIV-1 infection). Multiple parameters will be available from flow cytometric analyses and descriptive statistics including measures of central tendency will be compiled as tables. **The primary analytic endpoint of the study** was the SARS-CoV2-specific T cell profile depending on disease severity. All other variables were to be handled as exploratory endpoints. In the event the first wave of infection rapidly declined towards the end of the period outlined above the decision to close recruitment to the study was therefore made on pragmatic grounds.

## Serological testing

This was performed by Elecsys® (Roche, Basel, Switzerland) Anti-SARS-CoV-2 immunoassay which detects antibodies (including IgG) to SARS-CoV-2. The following routine clinical assays were performed in the laboratories of the National Health Laboratory Service (NHLS): full blood count and automated differential cell count, HIV-1 ELISA, Microbiology including tuberculosis Gene Xpert nucleic acid amplification testing, HIV-1 viral load, peripheral blood CD4 percentage and absolute count, SARS-CoV-2 diagnostic PCR (Seegene, Roche, Gene Xpert), blood electrolytes, C-reactive protein, D-dimer, ferritin and lactate dehydrogenase. The NHLS via its quality assurance division provides proficiency testing to its own and other laboratories schemes and is certified to ISO/IEC 17043:2010 requirements.

## Pseudovirus neutralization assay

Patient plasma was evaluated for SARS-CoV-2 neutralising activity using a pseudovirus neutralization assay. Single-cycle SARS-CoV-2 pseudovirions, based on the HIV backbone, were generated by co-transfection of plasmids pNL4-3.Luc.R-.E- (aidsreagent #3418) and pcDNA3.3-SARS-CoV-2-spike Δ18 into HEK-293TT cells[52]. Cell culture supernatants containing the virions were harvested 3 days post transfection and incubated with heat-inactivated plasma at 5-fold serial dilutions for 60 min at 37 °C. Plasma/pseudovirus mixtures were then used for transfection of HEK-293T cells stably expressing the ACE2 receptor[53]. Cells were lysed 3 days post infection using the Cell Culture Lysis Reagent (Promega) and assessed for luciferase activity using a GloMax® Explorer Multimode Microplate Reader (Promega) together with the Luciferase assay system (Promega). The half maximal inhibitory dilution (ID50) of each tested plasma sample was calculated in GraphPad Prism using a Non-linear regression[54].

## T cell stimulation by SARS-CoV-2 peptides

The assessment of the percentage of CD4 and CD8 T cells was performed on cryopreserved fixed whole blood as described[55]. Briefly bood was collected in sodium heparin tubes and processed within 3 h of collection. We adapted this assay to detect SARS-CoV-2 specific T cells using synthetic SARS-CoV-2 PepTivator peptides (Miltenyi Biotec, Surrey, UK), consisting of 15-mer sequences with 11 amino acid overlap covering the immunodominant parts of the spike (S) protein, and the complete sequence of the nucleocapsid (N) and membrane (M) proteins[55]. All peptides were combined in a single pool and used at a final concentration of 1 µg/ml. Briefly, 400 µl whole blood was stimulated with the SARS-CoV-2 S, N and M protein peptide pool at 37 °C for 5 h in the presence of co-stimulatory antibodies against CD28 (clone 28.2) and CD49d (clone L25) (1 µg/ml each; BD Biosciences, San Jose, CA, USA) and Brefeldin-A (10 µg/ml, Sigma-Aldrich, St Louis, MO, USA). Unstimulated blood was incubated with co-stimulatory antibodies, Brefeldin-A and an equimolar amount of DMSO. Red blood cell lysis and white cell fixation was performed in a single step using a Transcription Factor Fixation buffer (eBioscience, San Diego, CA, USA) for 20 min. Cells were then cryopreserved in freezing media (50% fetal bovine serum, 40% RPMI and 10% dimethyl sulfoxide) and stored in liquid nitrogen until batched analysis.

## Flow cytometry

Cell staining was performed on cryopreserved cells that were thawed, washed and permeabilized with a Transcription Factor perm/wash buffer (eBioscience). Cells were then stained at room temperature for 45 min with antibodies for Purified NA/LE mouse anti-human CD28 (clone 28.2), BD Pharmingen, Cat# 555725, RRID:AB_2130052, dilution: 1/1000; purified NA/LE mouse anti-human CD49d (clone L25), BD Pharmingen, Cat# 555501, RRID:AB_396068, dilution: 1/1000; CD3 BV650 (clone: OKT3), Biolegend, Cat# 317324, RRID:AB_11126748, dilution: 1/250; CD4 BV785 (clone: OKT4), Biolegend, Cat# 317441, RRID:AB_2561365, dilution: 1/83; CD8 BV510 (clone: RPA-8), Biolegend, Cat# 301048; RRID:AB_2561942, dilution 1/36; CD45RA Alexa 488 (clone: HI100), Biolegend, Cat# 304114, RRID:AB_528816, dilution: 1/50; CD27 PE-Cy5 (clone: 1A4CD27, Beckman Coulter), cat# 6607107, RRID:AB_10641617, dilution: 1/50; CD38 APC (clone: HIT2) BD Biosciences, Cat# 555462, RRID:AB_398599, dilution: 1/8; HLA-DR BV605 (clone L243), Biolegend, Cat# 307639, RRID:AB_11219187, dilution: 1/100; Ki67 PerCP-Cy5.5 (clone B56), BD Biosciences, Cat# 561284, RRID:AB_10611574, dilution: 1/125; PD-1 PE (Clone: J105) eBioscience), Cat# 12-2799-42, RRID:AB_11042478, dilution: 1/100; Granzyme B (GrB) BV421 (clone: BG11, BD Biosciences), Cat# 563389, RRID:AB_2738175, dilution: 1/167; IFN-γ BV711 (clone: 4 S.B3), BioLegend Cat# 502540, RRID:AB_2563506, dilution: 1/71; TNF-α PE-Cy7 (clone: MAB11) Biolegend, Cat# 502930, RRID:AB_2204079, dilution: 1/250; IL-2 PE/Dazzle 594 (clone: MQ1-17H12), Biolegend, Cat# 500344, RRID:AB_2564091, dilution: 1/71. Samples were acquired on a BD LSR-II and analyzed using FlowJo (v9.9.6, FlowJo LCC, Ashland, OR, USA). A positive response was defined as any cytokine response that was at least twice the background of unstimulated cells. To define the phenotype of SARS-CoV-2-specific CD4 T cells, a cut-off of 20 events was used.

## Radiographic evaluation

Chest radiographs were reported blind to clinical status by an experienced radiologist (QS-H). The BSTI coding system was employed to report chest radiographs as described[56] and the Brixia severity score was also calculated as described[35]. Posteroanterior chest radiographs were assessed for the total percentage of the lung fields unaffected by any visible pathology. Thus, in the COVID-19 group this score quantified the percentage of normal lung that was not visibly affected by known features of COVID-19 pneumonia on the radiograph. In those with tuberculosis, or in NC with other respiratory infections this score similarly quantified the percentage of normal lung, not visibly affected by the relevant pathology on the radiograph. Those with a normal chest radiograph would thus score 100%.

## Statistical methods

Analyses were conducted in Prism 8 (GraphPad Software, San Diego, CA). The normality of data was assessed by a Shapiro-Wilk test. Descriptive statistics were calculated and presented as percentage or median and IQR. Unpaired normally distributed variables were compared by the student's unpaired t test and non-parametric comparisons by the Kruskal-Wallis test. Corresponding assessment of non-parametric correlation was by the Spearman method.

## Reporting summary

Further information on research design is available in the Nature Portfolio Reporting Summary linked to this article.

# Data availability

Requests for clinical metadata will be reviewed by the HIATUS team, Wellcome Centre for Infectious Diseases Research in Africa to determine if the request is subject to confidentiality and data protection obligations. Data that can be shared will be released via a data sharing agreement. Source data are provided with this paper.

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

## Acknowledgements

We thank all study participants and their relatives for their willingness to be recruited into the study. We thank the management of Groote Schuur Hospital for permitting the study to take place and its staff for their practical assistance in its conduct. We are grateful for logistic support of Berenice Arendse in the Institute of Infectious Disease and Molecular Medicine at the University of Cape Town. HEK-293T cells stably expressing the ACE2 receptor were kindly provided by Huihui Mou, Department of Immunology and Microbiology, The Scripps Research Institute, Jupiter, FL, USA and the pcDNA3.3-SARS-CoV-2-spike Δ18 by Deli Huang, Department of Immunology and Microbiology, The Scripps Research Institute, La Jolla, CA, USA. This research was funded in whole, or in part, by Wellcome [104803, 203135, 222574]. For the purpose of Open Access, the author has applied a CC BY public copyright licence to any Author Accepted Manuscript version arising from this submission. RJW is supported by the Francis Crick Institute which receives its core funding from Cancer Research UK (FC0010218), the UK Medical Research Council (FC0010218), and Wellcome (FC0010218) and received additional support from the Rosetrees trust (M926) and the Medical Research Council of South Africa. GS is funded by EDCTP (TMA2018SF-2446). C.R. is supported by the EDCTP2 program of the European Union's Horizon 2020 program (TMA2017SF-1951-TB-SPEC to C.R.).

## Author contributions

R.J.W., Ed.B., C.S., M.C., A.K.C., K.A.W., A.G.D., R.P.-J.L. and C.R. conceived the study with input from Jd.V. and N.T. on ethical aspects. MSM, GM, NN and SW supervised clinical recruitment. Ed.B., R.D., C.S. recruited patients with logistic assistance from R.T.G. A.J. designed the clinical research forms and database and supervised data cleaning. B.W.A. conceptualised the percentage unaffected radiographic score. Q.S.H. performed the radiographic interpretation and scoring. F.A. coordinated the receipt and assay of laboratory specimens and M.H. was responsible for processing and reporting of samples in the NHLS analyses. G.S. performed pseudoviral neutralisation assays. Ed.B., C.S. and C.R. analyzed the data. All authors critically reviewed and provided input to the manuscript.

## Competing interests

All authors have completed the Unified Competing Interest form (available on request from the corresponding author) and declare: no financial relationships with any organisations that might have an interest in the submitted work in the previous three years, no other relationships or activities that could appear to have influenced the submitted work.

## Additional information

[1]Wellcome Centre for Infectious Diseases Research in Africa, Institute of Infectious Diseases and Molecular Medicine, University of Cape Town, Observatory 7925, Republic of South Africa. [2]Department of Medicine, University of Cape Town, Observatory 7925, Republic of South Africa. [3]Department of Infectious Diseases, Imperial College London, London W12 0NN, UK. [4]Department of Radiology, University of Cape Town, Observatory 7925, Republic of South Africa. [5]Department of Pathology, University of Cape Town, Observatory 7925, Republic of South Africa. [6]National Health Laboratory Service, Groote Schuur Complex, Department of Clinical Virology, Observatory 7925 Cape Town, Republic of South Africa. [7]Department of Integrated Biomedical Sciences, University of Cape Town, Observatory 7925, Republic of South Africa. [8]International Centre for Genetic Engineering and Biotechnology (ICGEB), Cape Town, South Africa. [9]Division of Pulmonology, Department of Medicine, Stellenbosch University and Tygerberg Hospital, Cape Town, Republic of South Africa. [10]The Francis Crick Institute, Midland Road, London NW1 1AT, UK. [11]Division of Life Sciences, University College London, London WC1E 6BT, UK. [12]The Walter and Eliza Hall Institute of Medical Research, Parkville Victoria 3052, Australia. [13]Health Impact Assessment unit, Western Cape Department of Health, Cape Town, Republic of South Africa. [14]Centre for Infectious Disease Epidemiology and Research, School of Public Health and Family Medicine, University of Cape Town, Observatory 7925, Republic of South Africa. [15]Division of Computational Biology, University of Cape Town, Observatory 7925, Republic of South Africa. [19]These authors contributed equally: Elsa du Bruyn, Cari Stek, Catherine Riou, Robert J. Wilkinson.
✉e-mail: cr.riou@uct.ac.za; Robert.Wilkinson@uct.ac.za

## HIATUS consortium

Fatimah Abrahams[1], Brian Allwood[9], Saalikha Aziz[1], Nonzwakazi Bangani[1], John Black[16], Melissa Blumenthal[2], Marise Bremer[1,16], Wendy Burgers[1,5], Maddalena Cerrone ®[1,3,10], Zandile Ciko[1], Anna K. Coussens[1,12], Remy Daroowala[1,3], Angharad G. Davis[1,10,11], Jantina de Vries ®[2], Elsa du Bruyn[1], Hanif G. Esmail[1], Rene T. Goliath ®[1], Siamon Gordon[17], Yolande X. R. Harley[1], Marvin Hsiao[1,5,7], Amanda Jackson[1], Rachel P.-J. Lai[3], Francisco Lakay[1], Fernando-Oneissi Martinez-Estrada[18], Graeme Meintjes[2], Marc S. Mendelson[1,2], Ntobeko Ntusi[1,2], Tari Papavarnavas[2], Alize Proust[10], Catherine Riou ®[1,5,19]✉, Sheena Ruzive[1], Qonita Said-Hartley[4], Georgia Schafer[1,8], Keboile Serole[1], Cari Stek[1,2,3,19], Nicki Tiffin ®[1,13,14,15], Sean Wasserman[1,2], Claire Whitaker[1], Katalin A. Wilkinson[1,10], Robert J. Wilkinson[1,2,3,10] & Kennedy Zvinairo[1]

[16]Livingstone Hospital, Gqeberha, Republic of South Africa. [17]Sir William Dunn School of Pathology, University of Oxford, Oxford, UK. [18]Department of Biochemical Sciences, University of Surrey, Guildford, UK.

