## [Peer Review File · Nature Communications]

Effects of tuberculosis and/or HIV-1 infection on COVID-19 presentation and immune response in AfricaReviewers' Comments:

Reviewer #1:

Remarks to the Author:

Summary:

In this manuscript Bruyn et al. report on an important cohort of COVID-19 patients co-infected with HIV and/or TB in an African setting. They show altered immunity against SARS-CoV-2 and higher mortality in the co-infected patients, especially those with coincident TB. They also show that HIV-infected patients recruited in the study appeared not to more likely to die than HIV-uninfected patients. Overall, this is a very rich cohort and dataset that would be of huge benefit to the field, however, the way the manuscript is written dampens my enthusiasm.

Major comments:

1. The manuscript could benefit with more informative sub-headings in the results section, and summary sentences at the end of each result sub-heading. This will help the reader easily follow the story, in its current state it will take a lot of effort on the part of the reader to figure out what is the main point.
2. The manuscript has a lot of Supplementary Figures, which itself is not a problem but how they are being used throughout the manuscript is confusing. For example, the authors beginning with describing Supplementary Figures (Page 6 Line 131, 150; Page 7 Line 152, 156) before the main figures (Page 7 Line 160, 161, 167). Supplementary Figures should be treated as supporting data and only referenced where necessary.
3. Page 6 Line 127-130: The authors should include p values. They also need to state the comparator group.
4. Page 12 Line 312-319: This paragraph is overstating the study findings. There is no clearly evidence that the age structure has not contributed to mitigating the COVID-19 pandemic in Africa. There is a lot of published literature with reported data reflecting a markedly less severe COVID-19 in Africa. This has not been proven otherwise.
 - Tessema, S.K., Nkengasong, J.N. Understanding COVID-19 in Africa. *Nat Rev Immunol*21, 469–470 (2021). <https://doi.org/10.1038/s41577-021-00579-y>
 - Uyoga et al. Seroprevalence of anti-SARS-CoV-2 IgG antibodies in Kenyan blood donorS. *Science*. 2021 Jan 1;371(6524):79-82. <https://www.science.org/doi/10.1126/science.abe1916>
5. The authors should provide or discuss pre-COVID-19 pandemic mortality rates in their study setting of the profile of patients recruited into the study. The HIV infected individuals seemed to have been severely immunocompromised and mortality in such patient groups with/without TB is very high in other settings.
6. The authors should include a paragraph in the discussion on the potential limitations of the study.
7. Figure 5A, B & E: What is the basis for comparing CD4 to CD8 T cell responses? Could be this be stated in the results section? Did they expect to find similar responses? It would seem obvious that they would differ. Why did they not compare CD4 or CD8 T cell alone among the WHO severity groups?
8. The abstract is not clear and concise. It would benefit with some re-writing.

Minor comments:

1. Page 3 Line 43: "COVID-19 infection" should read "SARS-CoV-2 infection". COVID-19 is a disease
2. Page 4 Line 78 & Page 12 Line 310: The link should be in the reference section and should only be referenced with a number like the rest of the references
3. Page 6 Line 127: What does days of illness to blood sampling mean?
4. Page 6 Line 134: Sentences shouldn't start with numbers. e.g. "46.2% COVID-19..."; instead can be rewritten as "Forty six percent of COVID-19...". This should be done throughout the manuscript
5. Figures need to be referenced as they appear. e.g. Line 163 (Figure 1C) and Line 161 (Figure 1D and E). Figure 1C should be referenced before Figure 1D and E
6. Page 8 Line 193: Why did authors assess lymphopenia by flow cytometric analysis?
7. Figure 4B-C: The authors should replace "Lymph" with "Lymphocyte"

Reviewer #2:

Remarks to the Author:

This work by du Bruyn et al. investigates SARS-CoV-2 outcome in hospitalized participants with HIV, TB, and HIV/TB co-infection and compares them to non-SARS-CoV-2 infected participants hospitalized for other reasons (NC). They found that: 1) In SARS-CoV-2/TB+ participants, clinical features could be dominated by either SARS-CoV-2 or tuberculosis, the degree of lymphopenia was greater, and D-dimer and ferritin was elevated. 2) SARS-CoV-2/HIV+ participants with the lowest CD4 percentage showed reduced anti-SARS-CoV-2 antibodies. 3) In SARS-CoV-2+/TB+/HIV+ participants, there were no SARS-CoV-2 specific CD8 T cell responses. Some of the conclusions were that the immune response to SARS-CoV-2 is adversely affected by HIV and TB co-infection, that HIV and TB co-infection is not uncommon in SARS-CoV-2 in TB endemic areas, and that SARS-CoV-2 may co-exist in typical presentations of TB and perhaps be involved in reactivation.

The strengths of the paper are the cohort which allows the authors to investigate the effects of HIV, TB and HIV/TB co-infection on SARS-CoV-2 infection, and the result that the CD8 SARS-CoV-2 response is compromised in SARS-CoV-2+/TB+/HIV+ participants. The weaknesses are that the results are observational with some having an unclear message or repeating what is well known in SARS-CoV-2 infection without showing strong effects of HIV, TB or HIV/TB co-infection, and overlap with a previous publication of the authors in JCI. Furthermore, the comparison between SARS-CoV-2+ and the NC groups is confounded by the higher disease severity in the SARS-CoV-2 participants. Given that the cohort is not a random sample, inferences about how common SARS-CoV-2/TB/HIV co-infection is should be qualified. These points are detailed below:

1) The cohort, both the SARS-CoV-2 and NC arms, seems to enroll hospitalized participants at Groote Schuur Hospital.

A) Given the above, why are 7 of the NC participants listed as WHO ordinal scale 2 (not hospitalized, limitation on activities)?

B) About half of the SARS-CoV-2 group also are categorized as severe by the WHO ordinal scale, with none of the NC being severe. Differences in C-Reactive protein, D-dimer, ferritin, and lactate dehydrogenase as shown in Figure 3 may be entirely related to disease severity and may not be specific to SARS-CoV-2 infection.

C) The relatively high fraction of SARS-CoV-2+/TB+/HIV+ participants may be because: i) this is common in the population, or ii) the co-infection causes a worse SARS-CoV-2 outcome leading to hospitalization. The conclusion that SARS-CoV-2+/TB+/HIV+ is not uncommon should be qualified to happen in hospital admissions.

2) The meaning of the results in Fig 1F is unclear: either a combination of HIV/TB in conjunction with SARS-CoV-2 leads to lower lung pathology, or SARS-CoV-2+/TB+/HIV+ participants have more severe Covid-19 disease and so are more likely to be admitted to hospital with less lung pathology. Analysis of additional symptoms would be helpful to understand which of the two applies.

3) I could not find a figure comparing the WHO ordinal severity score for SARS-CoV-2+, SARS-CoV-2+/HIV+, SARS-CoV-2+/TB+, and SARS-CoV-2+/HIV+/TB+. This should be presented up front.

4) The message of Fig 2 is unclear. For example, lymphopenia is known to be associated with more severe Covid-19 as well as other diseases, and neutrophils are known to go up with severity. The results do not clearly show that SARS-CoV-2+/TB+/HIV+ (or the other infection groups) have markers significantly associated with higher disease severity relative to SARS-CoV-2 infection alone.

5) Nucleocapsid antibodies do not indicate whether there is an effective neutralization response and do not indicate functional immune differences. The results of Fig 4 should be repeated using a

neutralization assay with participant plasma/serum to compare SARS-CoV-2+, SARS-CoV-2+/HIV+, SARS-CoV-2+/TB+, and SARS-CoV-2+/HIV+/TB+ neutralization capacity, as well as the effect of CD4 T cell count. CD4 T cell count should also be presented as absolute numbers (more often used in the HIV field), not only % of total.

6) Fig 4C is identical as far as I can see to Fig 6F in a previous publication of the authors in JCI (see <https://www.jci.org/articles/view/149125/figure/6>). This should be clearly stated. The statement on lines 304-306: "Of potentially greater significance, in terms of longer lasting immunity in convalescence, there was a direct relationship between the peripheral CD4 percentage and acute antibody levels." Should be referenced with J Clin Invest. 2021;131(12):e149125.

Minor points

1) Fig 5D: The category labels should be on the graph and the meaning of the colored squares in the figure legend, otherwise difficult to understand.

2) Lines 333-335: "Thus, in environments in which tuberculosis is common, clinical suspicion of reactivation and also that SARS-CoV-2 may co-exist in typical presentations of tuberculosis are the most important clinical lessons from our work." This is unclear. If the implication is that there is reactivation of TB with SARS-CoV-2 infection, this cannot be determined from the data presented in this study.

Reviewer #3:

Remarks to the Author:

Key results

This is a useful study of the clinical presentation and laboratory features of COVID-19 in a single centre hospital setting, in a country with high burden of HIV, TB and non-communicable disease and obesity. The following findings were of interest: HIV and TB were not uncommon co-morbidities in COVID-19 patients. Mortality in COVID/TB co-infected patients was 40%, in COVID/HIV co-infected patients was 19%, in COVID/HIV/TB co-infected was 38% (mortality not increased among COVID/HIV co-infected). In patients co-infected with COVID/TB, clinical presentation was typical of either disease; and lymphopenia and inflammatory markers were more elevated. Only LDH was associated with disease severity. Radiographic presentation was typical of COVID-19 except in a small number of COVID/TB co-infected patients where TB findings were predominant. In patients co-infected with COVID/HIV/TB, the immune response was adversely affected. The study revealed a number of COVID-19 patients with unsuspected TB and suggest SARS-CoV-2 immune effects, combined with corticosteroid therapy may contribute to reactivation of subclinical or latent TB.

Data and methodology

The main body of the manuscript does not contain a Methods section, this is available only in the online methods section. The abstract also does not provide any detail on methods. This is a major flaw of the paper, as a reader has little information on the selection of SARS-CoV-2 cases and NC controls. The sample size has not been explained.

Also, the results section makes reference to other control groups (Line 176: pre-pandemic controls; and Line 198: prior study controls from community health clinic). These controls were not described even in the online methods. The manuscript requires greater clarity in the description of the study population and how they were recruited.

Analytical approach

Appropriate descriptive analysis was employed.

Clarity and context

Abstract would be more accessible if it has sub-sections (introduction, Methods, Results, Conclusions).

There is no methods presented in the body of the manuscript, only an online Methods section.

The results section and figures are very dense with content but the authors have tried their best to

make the findings logical to follow. However some sentences are very complex (eg line 163) and can be made more readable. Otherwise the paper is clear, well-structured and accurately presented and there is sufficient context and consideration of previous work. Tables, figures and supplementary material are presented well. The paper does have an ability to speak to key audiences.

References-

The manuscript does not adequately cite literature on risk for COVID-19 mortality associated with HIV and TB. There are larger population cohorts (SA and UK) and a national hospital surveillance analysis (SA) with more recent evidence of increased risk of mortality for HIV and TB. There have been a few more studies published or in preprint that the authors could also reference. Also see WHO report on Covid and HIV, and latest ISARIC Covid hospitalisation analysis.

Line 96: the authors quote older small single centre studies, there are population cohorts from UK and SA, and hospitalised cohorts in SA that show poorer outcome (see also Jassat, Lancet HIV)

Line 101: not much literature cited on TB and Covid mortality (see Jassat, Lancet HIV)

Validity

On the whole the study is robust in its interpretation of clinical presentation, laboratory and radiological investigations. I do however have some concerns around study limitations that I believe that editor should consider in making a decision on the manuscript's publication.

The study discusses some limitations (not designed to assess risk factors for mortality, not included formal controls, potential confounder of steroid therapy), there are other limitations not discussed which may represent bias in the analysis.

1. The sample of patients is very small and the sample size has not been well described, and neither have the comparator control groups from prior studies been described in terms of selection.

2. There was a difference between Covid and NC cases in severity. None of the NC patients were categorised on the higher end of the WHO ordinal severity score 5, 6 or 7.

3. The NC cases were not recruited at the same time as the COVID cases but after (Line 355).

Sampling prioritised positive cases first then PUIs after. Study from South Africa has shown that in-hospital mortality was highest during the peaks of the waves when hospital capacity was overwhelmed (Jassat, Lancet Global Health). Not recruiting the comparison group at the same time has potential to bias the analysis, as there could be differences in patient severity and quality of care for those who present during the surge and those who present after.

4. The study concurs with other studies that show HIV is not associated with worse outcomes.

However this is a small sample single centre study, similar to other studies quoted in the manuscript.

A more extensive reading of the literature especially more recent literature suggests an association for HIV and TB with Covid mortality. See WHO report on Covid and HIV, and latest ISARIC Covid hospitalisation analysis.

5. The issue of steroid treatment has been mentioned as a potential confounder, but this would seriously limit the conclusions drawn around inflammatory markers.

The quality of the data

The manuscript employed appropriate methods and analysis, presented findings in appropriate detail, and interpreted findings carefully. My concerns are regarding the small sample size, the description of the study population, particularly controls, differences between covid and NC cases, and potential bias around not recruiting COVID and NC cases at the same time.

The level of support for the conclusions

The main study claims are justified by the data and analytic methods used. However the small sample size and bias may be problematic. The interpretation of no association of HIV with mortality may not be supported due to small sample size and conflicting evidence from large population and hospital cohorts.

The potential significance of the results

The findings do contribute to broader understanding of HIV, TB co-infection with SARS-CoV-2, and their clinical presentation and outcome. The evidence and arguments presented support advancement of COVID-19 understanding especially for high burden HIV and TB countries, and for Africa where burden of HIV and TB is high and diagnosis of SARS-CoV-2 is likely under-ascertained. It also provides caution in these settings for maintaining clinical vigilance for undetected TB.

Your expertise

The interpretation of immune response data is outside the scope of my expertise, and I was unable to assess fully.

Suggested improvements

I have made the following minor suggestions that could help strengthen the work and make it suitable for publication in the journal.

- Suggest revise the title to make it more concise and explanatory
- Use HIV or PLHIV rather than "HIV-1"
- Use sub headings in abstract
- Abstract line 45 states this is an observational case cohort; the abstract does not adequately describe the methods and does not state that the patients are sampled from hospitalised patients
- Line 87: proportion who died is 2.8% not 28%, check this calculation

Reviewer #4:

Remarks to the Author:

Chest radiographs at enrollment were evaluated by three different scoring systems: Brixia, BSTI and percentage of unaffected lung

Only one of the authors is a radiologist. It would be important to know if this author was involved in the interpretation of the images. It would be also important to know if more authors interpreted the images and the degree of agreement between different readers and how the agreement was achieved

BRIXIA (ref. # 32) refers to a chest radiography score system that examines severity of SARS-CoV-2 disease with age and sex.

A more recent paper by the same author (PMID 33263159) describes that Brixia score correlates strongly with disease severity and outcome

BSTI score (BTSICXR) classifies radiographies as classical, indeterminate, and not covid typical images based on imaging patterns

Reference # 33 refers to the ability of "radiographers" to adequately use and apply the BSTI classification system when reporting COVID-19 chest radiographs. This reference may not be adequate for the purpose of this manuscript

"Percentage of unaffected lung" is not explained in the methods. A reference is not given

Figure 1, A1. It is not clear why CT images are relevant in this manuscript

"Bibasal wedge shape opacities" are not always indicative of pulmonary emboli. This is a CT angiogram which should be able to determine pulmonary artery filling defects, which are more indicative of pulmonary emboli

Fig B1 A. "diffuse bilateral shadowing:" is not an appropriate radiology term.

Fig B2. This lesion described is not clearly a cavitory lesion and the upper arrow does not clearly show enlarged subcarinal or precranal lymph nodes

REVIEWER COMMENTS

Reviewer #1 (Remarks to the Author):

Summary:

In this manuscript Bruyn et al. report on an important cohort of COVID-19 patients co-infected with HIV and/or TB in an African setting. They show altered immunity against SARS-CoV-2 and higher mortality in the co-infected patients, especially those with coincident TB. They also show that HIV-infected patients recruited in the study appeared not to more likely to die than HIV-uninfected patients. Overall, this is a very rich cohort and dataset that would be of huge benefit to the field, however, the way the manuscript is written dampens my enthusiasm.

We are grateful the reviewer finds this a ‘very rich cohort’ and have made revisions in responses to the specific points below.

Major comments:

1. The manuscript could benefit with more informative sub-headings in the results section, and summary sentences at the end of each result sub-heading. This will help the reader easily follow the story, in its current state it will take a lot of effort on the part of the reader to figure out what is the main point.

We accept the manuscript is complex and have revised, and also added, subheadings in the results section. Thus the following sections are now included in the results (new subheadings in red)

Participants included in the analysis

Mortality due to COVID-19

Radiographic features

Clinical features of COVID-19 in participants with co-incident tuberculosis and/or HIV-1 infection

Further analysis of lymphopenia by flow cytometric analysis

Serum biomarkers

SARS-CoV-2 specific antibody response

SARS-CoV-2 specific CD8⁺ T cell response

2. The manuscript has a lot of Supplementary Figures, which itself is not a problem but how they are being used throughout the manuscript is confusing. For example, the authors beginning with describing Supplementary Figures (Page 6 Line 131, 150; Page 7 Line 152, 156) before the main figures (Page 7 Line 160, 161, 167). Supplementary Figures should be treated as supporting data and only referenced where necessary.

We acknowledge there is a lot of data as the reviewer has pointed out. We do begin the results section by describing the participants which is a table (1) in the main set of display pieces. Due to limits on the number of display items we had to decide what was crucial in the narrative and what was supplementary. We do acknowledge this results in several references to supplementary material early in the results section, but we can only cite in the order in which it appears.

3. Page 6 Line 127-130: The authors should include p values. They also need to state the comparator group.

The text of lines 127-130 presently states ‘HIV-1 co-infected NC had a lower CD4 count (median: 18 cells/mm³ [IQR: 7-102] versus 132 [51-315], p=0.028) and higher median HIV-1 viral load (median: 5.36 log₁₀ HIV RNA copies/ml [IQR: 2.49-5.21] versus <1.3 [<1.3-4.14], p=0.0005)’. So it appears we have specified the comparator group and p values. We apologise if there is some confusion here and seek guidance if we are in error.

4. Page 12 Line 312-319: This paragraph is overstating the study findings. There is no clearly evidence that the age structure has not contributed to mitigating the COVID-19 pandemic in Africa. There is a lot of published literature with reported data reflecting a markedly less severe COVID-19 in Africa. This has not been proven otherwise.

- Tessema, S.K., Nkengasong, J.N. Understanding COVID-19 in Africa. *Nat Rev Immunol* 21, 469–470 (2021). <https://doi.org/10.1038/s41577-021-00579-y>
- Uyoga et al. Seroprevalence of anti-SARS-CoV-2 IgG antibodies in Kenyan blood donorS. *Science*. 2021 Jan 1;371(6524):79-82. <https://www.science.org/doi/10.1126/science.abe1916>

We acknowledge this point and deleted the introductory sentence and now simply state ‘Our study confirms that a wide range of now well-recognised co-morbidities (most commonly hypertension, type 2 diabetes mellitus and obesity) that associate with COVID-19 severity and death were present in the bulk of our cases.

5. The authors should provide or discuss pre-COVID-19 pandemic mortality rates in their study setting of the profile of patients recruited into the study. The HIV infected individuals seemed to have been severely immunocompromised and mortality in such patient groups with/without TB is very high in other settings.

We accept this point but for reasons the reviewer and other reviewers have pointed out this cannot be addressed in this observational study. We do point out in the introduction that the South African Medical Research Council estimates 326,280 excess deaths nationally since 3rd May 2020. Work on this dataset will be ongoing and may yield estimates of excess death in specific subgroups in future. We also point out that the first and all subsequent analyses at the population level in the Western Cape have indicated active tuberculosis and HIV-1 (particularly in those highly immunosuppressed) are independent risk factors for COVID-19 mortality.

6. The authors should include a paragraph in the discussion on the potential limitations of the study.

We did discuss limitations albeit admit these were not compiled into a single paragraph. Because of word limit constraints we have elected an approach that now specifically acknowledges where a limitation exists e.g. Paragraph 2 of the discussion now begins ‘Our study benefitted inclusion of participants admitted to the same hospital at the same time as the surge in COVID-19 admissions who were subsequently assigned alternative diagnoses with negative virological and serological tests for SARS-CoV-2. **A limitation is that this is not however a formal control group (or reported as such) and we cannot exclude, under the circumstances, that such tests were false negative.**’

Again in the first sentence of paragraph 5: ‘Another study limitation is the analysis was not designed or powered to reproduce the previously documented relationships between HIV-1 co-infection and tuberculosis and risk of death from COVID-19’.

We have also made a number of further acknowledgements of limitations as a consequence of the points of other reviewers.

7. Figure 5A, B & E: What is the basis for comparing CD4 to CD8 T cell responses? Could be this be stated in the results section? Did they expect to find similar responses? It would seem obvious that they would differ. Why did they not compare CD4 or CD8 T cell alone among the WHO severity groups?

The reason for this comparison is to show that CD8 responses were less frequent but, where present, of equal magnitude to the CD4 response. The purpose of panel 5E is to show the CD8 response has a distinct phenotype when compared to CD4, being more likely to be HLA-DR positive and Granzyme B positive. We do reports this in the text as follows, ‘found evidence of terminal effector differentiation in CD8 cells which were more frequently positive for HLA-DR, CD38, Granzyme B and PD-1 than CD4 cells (Figure 5E).’

We have previously reported the CD4 response in relation to WHO severity scale (Riou *et al.* J Clin Invest 2021) and so do not repeat that analysis in this manuscript. However we do show the CD8 response in relationship to severity in panel D of Figure 5 which indicates there is a trend towards increased percentage of Interferon-gamma single positive cells as severity increases, with a corresponding decrease in the percentage of cells that are polyfunctional.

8. The abstract is not clear and concise. It would benefit with some re-writing.

We have tried to state the background, purpose, main findings, and interpretation. If the reviewer and/or editor can provide specific guidance what should be made more clear we would be pleased to do this.

Minor comments:

1. Page 3 Line 43: “COVID-19 infection” should read “SARS-CoV-2 infection”. COVID-19 is a disease

Acknowledged and corrected.

2. Page 4 Line 78 & Page 12 Line 310: The link should be in the reference section and should only be referenced with a number like the rest of the references

Amended as requested.

3. Page 6 Line 127: What does days of illness to blood sampling mean?

The number of days the participant was ill prior to blood sampling.

4. Page 6 Line 134: Sentences shouldn’t start with numbers. e.g. “46.2% COVID-19...”; instead can be rewritten as “Forty six percent of COVID-19...”. This should be done throughout the manuscript

Amended as requested throughout

5. *Figures need to be referenced as they appear. e.g. Line 163 (Figure 1C) and Line 161 (Figure 1D and E). Figure 1C should be referenced before Figure 1D and E*

We acknowledge there is a minor deviation from citing convention here. This was done because of the shape of figures within the panels. If the editor finds this critical, we will reorganise the figures.

6. *Page 8 Line 193: Why did authors assess lymphopenia by flow cytometric analysis?*

The routine coulter counter only yields absolute and percent lymphocytes and thus no distinction between CD3, CD19, CD4 and CD8 positive subsets can be made.

7. *Figure 4B-C: The authors should replace “Lymph” with “Lymphocyte”*

Amended as requested

Reviewer #2 (Remarks to the Author):

This work by du Bruyn et al. investigates SARS-CoV-2 outcome in hospitalized participants with HIV, TB, and HIV/TB co-infection and compares them to non-SARS-CoV-2 infected participants hospitalized for other reasons (NC). They found that: 1) In SARS-CoV-2/TB+ participants, clinical features could be dominated by either SARS-CoV-2 or tuberculosis, the degree of lymphopenia was greater, and D-dimer and ferritin was elevated. 2) SARS-CoV-2/HIV+ participants with the lowest CD4 percentage showed reduced anti-SARS-CoV-2 antibodies. 3) In SARS-CoV-2+/TB+/HIV+ participants, there were no SARS-CoV-2 specific CD8 T cell responses. Some of the conclusions were that the immune response to SARS-CoV-2 is adversely affected by HIV and TB co-infection, that HIV and TB co-infection is not uncommon in SARS-CoV-2 in TB endemic areas, and that SARS-CoV-2 may co-exist in typical presentations of TB and perhaps be involved in reactivation.

The strengths of the paper are the cohort which allows the authors to investigate the effects of HIV, TB and HIV/TB co-infection on SARS-CoV-2 infection, and the result that the CD8 SARS-CoV-2 response is compromised in SARS-CoV-2+/TB+/HIV+ participants.

We thank the reviewer for these positive observations.

The weaknesses are that the results are observational with some having an unclear message or repeating what is well known in SARS-CoV-2 infection without showing strong effects of HIV, TB or HIV/TB co-infection, and overlap with a previous publication of the authors in JCI. Furthermore, the comparison between SARS-CoV-2+ and the NC groups is confounded by the higher disease severity in the SARS-CoV-2 participants. Given that the cohort is not a random sample, inferences about how common SARS-CoV-2/TB/HIV co-infection is should be qualified. These points are detailed below:

1) The cohort, both the SARS-CoV-2 and NC arms, seems to enroll hospitalized participants at Groote Schuur Hospital.

A) Given the above, why are 7 of the NC participants listed as WHO ordinal scale 2 (not hospitalized, limitation on activities)?

Seven patients are included who were not hospitalised following outpatient assessment at Groote Schuur. This has been clarified in the legend to Table 1.

B) About half of the SARS-CoV-2 group also are categorized as severe by the WHO ordinal scale, with none of the NC being severe. Differences in C-Reactive protein, D-dimer, ferritin, and lactate dehydrogenase as shown in Figure 3 may be entirely related to disease severity and may not be specific to SARS-CoV-2 infection.

We agree this is a limitation and indeed it is a point we try to make later in the same discussion. Perhaps, on reflection, we should not have used the WHO ordinal classification at all for the NC group because the ordinal score is heavily weighted by the need for, and type of, respiratory support. What we have elected is to reflect the reviewer's point as a limitation in the discussion as follows, 'In addition their ordinal score (which is dominated by consideration of the need for and type of respiratory support) was lower and it could be argued differences in biomarkers may related to severity and may not be specific to SARS-CoV-2 infection'

C) The relatively high fraction of SARS-CoV-2+/TB+/HIV+ participants may be because: i) this is common in the population, or ii) the co-infection causes a worse SARS-CoV-2 outcome leading to hospitalization. The conclusion that SARS-CoV-2+/TB+/HIV+ is not uncommon should be qualified to happen in hospital admissions.

Agreed and qualified in the amended abstract as follows, 'In high incidence setting, tuberculosis is a common co-morbidity in patients admitted to hospital with COVID-19....

2) The meaning of the results in Fig 1F is unclear: either a combination of HIV/TB in conjunction with SARS-CoV-2 leads to lower lung pathology, or SARS-CoV-2+/TB+/HIV+ participants have more severe Covid-19 disease and so are more likely to be admitted to hospital with less lung pathology. Analysis of additional symptoms would be helpful to understand which of the two applies.

We agree either hypothesis could be true and relates to the next point of the reviewer.

3) I could not find a figure comparing the WHO ordinal severity score for SARS-CoV-2+, SARS-CoV-2+/HIV+, SARS-CoV-2+/TB+, and SARS-CoV-2+/HIV+/TB+. This should be presented up front.

We have done this and included it as Figure 1G. Interestingly HIV positive status was associated with a slightly, but significantly, lower Ordinal score irrespective of the presence of tuberculosis. We have reported this finding in results as follows: 'This impression was supported by analysis of WHO ordinal scale at presentation which was slightly but significantly lower than that of HIV-1 uninfected SARS-CoV-2 mono-infected patients (median 5, IQR 4-6) in the presence of HIV-1 alone (median 4, IQR 4-5, p= 0.038) and in HIV-1 and tuberculosis co-infected SARS-CoV-2 participants (median 4, IQR 3-4, p = 0.008)'

4) The message of Fig 2 is unclear. For example, lymphopenia is known to be associated with

more severe Covid-19 as well as other diseases, and neutrophils are known to go up with severity. The results do not clearly show that SARS-CoV-2+/TB+/HIV+ (or the other infection groups) have markers significantly associated with higher disease severity relative to SARS-CoV-2 infection alone.

We agree with respect to neutrophils no difference was detected. With respect to lymphocytes triply infected patients had significantly lower levels than those doubly infected by SARS-CoV-2 and HIV-1 and this is reported in the results.

5) Nucleocapsid antibodies do not indicate whether there is an effective neutralization response and so do not indicate functional immune differences. The results of Fig 4 should be repeated using a neutralization assay with participant plasma/serum to compare SARS-CoV-2+, SARS-CoV-2+/HIV+, SARS-CoV-2+/TB+, and SARS-CoV-2+/HIV+/TB+ neutralization capacity, as well as the effect of CD4 T cell count. CD4 T cell count should also be presented as absolute numbers (more often used in the HIV field), not only % of total.

We accept the criticism and performed pseudovirus neutralisation (described in methods) and, where available, related this to absolute CD4 count. We have reported these new results as follows. **‘We have previously reported positive correlation between the percentage of CD4 T cells and the antibody cut-off index with triply infected patients tending to show low levels of both parameters⁴². We therefore further investigated in a subset of 91 SARS-COV-2 positive (64 HIV-TB-, 27 HIV+TB-, 7 HIV-TB+, 7 HIV+TB+) for whom adequate sample remained available the serum neutralization of spike-containing SARS-CoV-2 pseudovirus. No significant difference in ability to neutralize SARS-CoV-2 pseudovirus between groups was observed although a minority not prescribed antiretroviral therapy (darker shading) tended to have lower ID₅₀ values Figure 4C). In subset of HIV-1 co-infected patients for whom the absolute CD4 count was available, there was no trend between neutralising antibody level and CD4 (p = 0.34) and patients with tuberculosis (pink) did not form a distinct subgroup (Figure 4D)’.**

6) Fig 4C is identical as far as I can see to Fig 6F in a previous publication of the authors in JCI (see <https://www.jci.org/articles/view/149125/figure/6>). This should be clearly stated. The statement on lines 304-306: “Of potentially greater significance, in terms of longer lasting immunity in convalescence, there was a direct relationship between the peripheral CD4 percentage and acute antibody levels.” Should be referenced with J Clin Invest. 2021;131(12):e149125.

Because of the reviewer’s suggestion to determine neutralising antibodies, which we accepted and have done, this figure has been extensively reworked and no longer includes panel C: instead we refer to our prior published observation as above. We have moderated the statement in the discussion to **‘There were relationships between the peripheral CD4 percentage (and a trend for absolute CD4) and acute antibody levels. Whilst this may reflect duration of COVID-19 illness and its severity (both tending to increase antibody levels), it remains a concern that both natural and vaccine-induced immune responses to SARS-CoV-2 may be less effective in persons immunosuppressed by HIV-1, especially in those not prescribed ART’.**

Minor points

1) Fig 5D: The category labels should be on the graph and the meaning of the colored squares in the figure legend, otherwise difficult to understand.

This key was on the pie chart, but we have duplicated it on the left side of the figure and hope this make the graph easier to understand.

2) Lines 333-335: “Thus, in environments in which tuberculosis is common, clinical suspicion of reactivation and also that SARS-CoV-2 may co-exist in typical presentations of tuberculosis are the most important clinical lessons from our work.” This is unclear. If the implication is that there is reactivation of TB with SARS-CoV-2 infection, this cannot be determined from the data presented in this study.

We have modified the text as follows ‘Thus, in environments in which tuberculosis is common, clinical suspicion of **its presence** and **also** that SARS-CoV-2 may also co-exist in typical presentations of tuberculosis are the most important clinical lessons from our work.

Reviewer #3 (Remarks to the Author):

Key results

This is a useful study of the clinical presentation and laboratory features of COVID-19 in a single centre hospital setting, in a country with high burden of HIV, TB and non-communicable disease and obesity. The following findings were of interest: HIV and TB were not uncommon co-morbidities in COVID-19 patients. Mortality in COVID/TB co-infected patients was 40%, in COVID/HIV co-infected patients was 19%, in COVID/HIV/TB co-infected was 38% (mortality not increased among COVID/HIV co-infected). In patients co-infected with COVID/TB, clinical presentation was typical of either disease; and lymphopenia and inflammatory markers were more elevated. Only LDH was associated with disease severity. Radiographic presentation was typical of COVID-19 except in a small number of COVID/TB co-infected patients where TB findings were predominant. In patients co-infected with COVID/HIV/TB, the immune response was adversely affected. The study revealed a number of COVID-19 patients with unsuspected TB and suggest SARS-CoV-2 immune effects, combined with corticosteroid therapy may contribute to reactivation of subclinical or latent TB.

We thank the reviewer for these positive observations.

Data and methodology

The main body of the manuscript does not contain a Methods section, this is available only in the online methods section. The abstract also does not provide any detail on methods. This is a major flaw of the paper, as a reader has little information on the selection of SARS-CoV-2 cases and NC controls.

This criticism reflects similar points by other reviewers. In terms of the detailed methods, we are following journal policy. In terms of the abstract, we have now broken this down into subheadings and tried to supply more details within the word limits allowed.

The sample size has not been explained.

The protocol stated a priori the intent was to include 120 confirmed COVID-19 patients (equal numbers with or without HIV-1 co-infection) and 40 COVID-19 negative participants (equal numbers with or without HIV-1 infection). Multiple parameters were available from flow cytometric analyses and descriptive statistics including measures of central tendency were compiled as tables. The primary analytic endpoint of the study was the SARS-CoV2-specific T cell profile depending on disease severity. All other variables were to be handled as exploratory endpoints. In the event the first wave of infection rapidly declined towards the end of the period outlined above the decision to close recruitment to the study was therefore made on pragmatic grounds. This is now stated in methods.

Also, the results section makes reference to other control groups (Line 176: pre-pandemic controls; and Line 198: prior study controls from community health clinic). These controls were not described even in the online methods. The manuscript requires greater clarity in the description of the study population and how they were recruited.

We regret any lack of clarity. We did refer to this in results as below, and also in the methods and in supplementary table 4.

To investigate lymphocyte differences from COVID-19 patients, additional values were obtained from a subset of 118/163 similar ambulant participants enrolled to a prior study of HIV-1 uninfected and infected persons with either immune evidence of tuberculosis sensitization but no symptoms or microbiologically confirmed pulmonary tuberculosis treated at a community health clinic (Extended Table 4 and ^{38,39}).

In the methods the following is stated. ‘For some analyses we included data from HIV-1 uninfected and infected persons with either immune evidence of tuberculosis sensitization but no symptoms (latent tuberculosis, LTBI) or microbiologically confirmed pulmonary tuberculosis who had been recruited to prior studies (HREC 050/2015) ^{38,39}. The details of these persons are presented in Extended Table 4.’

Analytical approach

Appropriate descriptive analysis was employed.

Clarity and context

Abstract would be more accessible if it has sub-sections (introduction, Methods, Results, Conclusions).

This has been done as outlined above. We have created subheadings Rationale, Methods, Results and Conclusions.

There is no methods presented in the body of the manuscript, only an online Methods section. The results section and figures are very dense with content but the authors have tried their best to make the findings logical to follow. However some sentences are very complex (eg line 163) and can be made more readable.

We are following journal policy with respect to methods. We have broken this sentence into two as follows: ‘**In HIV-1-co-infected, tuberculosis-positive, SARS-CoV-2 positive patients the Brixia score was lower than other patient groups. This was significant for both singly SARS-CoV-2 positive patients and HIV-1 uninfected, tuberculosis-positive, SARS-CoV-2**

positive patients (median score: 8 vs. 14 and 15.5, $p = 0.006$ and 0.038 , respectively, Figure 1F).

Otherwise the paper is clear, well-structured and accurately presented and there is sufficient context and consideration of previous work. Tables, figures and supplementary material are presented well. The paper does have an ability to speak to key audiences.

We again thank the reviewer for these encouraging comments.

References-

The manuscript does not adequately cite literature on risk for COVID-19 mortality associated with HIV and TB. There are larger population cohorts (SA and UK) and a national hospital surveillance analysis (SA) with more recent evidence of increased risk of mortality for HIV and TB. There have been a few more studies published or in preprint that the authors could also reference. Also see WHO report on Covid and HIV, and latest ISARIC Covid hospitalisation analysis.

Line 96: the authors quote older small single centre studies, there are population cohorts from UK and SA, and hospitalised cohorts in SA that show poorer outcome (see also Jassat, Lancet HIV)

Line 101: not much literature cited on TB and Covid mortality (see Jassat, Lancet HIV)

We apologise but in mitigation point out at the time of submission (August 2021) that these analyses were not available. We have now included additional references in the introduction including those suggested by the reviewer. Additional text has been introduced as follows
These analyses were later confirmed in a very large South African national study in which associated were HIV infection (aOR 1.34), past tuberculosis (1.26), and current tuberculosis (1.42)¹²

AND

Amongst patients admitted to hospital in the UK cumulative day-28 mortality from COVID-19 was similar HIV-1 positive versus negative groups (26.7% vs. 32.1), but in those under 60 years of age HIV-positive status was associated with increased mortality (21.3% vs. 9.6%)²⁸.

Validity

On the whole the study is robust in its interpretation of clinical presentation, laboratory and radiological investigations. I do however have some concerns around study limitations that I believe that editor should consider in making a decision on the manuscript's publication. The study discusses some limitations (not designed to assess risk factors for mortality, not included formal controls, potential confounder of steroid therapy), there are other limitations not discussed which may represent bias in the analysis.

1. The sample of patients is very small and the sample size has not been well described, and neither have the comparator control groups from prior studies been described in terms of selection.

See comments above on sample size

2. There was a difference between Covid and NC cases in severity. None of the NC patients were categorised on the higher end of the WHO ordinal severity score 5, 6 or 7.

This is similar to a comment of reviewer 2. We agree this is a limitation and indeed it is a point we try to make later in the same discussion. Perhaps, on reflection, we should not have used the WHO ordinal classification at all for the NC group because the ordinal score is heavily weighted by the need for, and type of, respiratory support. What we have elected is to reflect the reviewer's point as a limitation in the discussion as follows, 'In addition their ordinal score (which is dominated by consideration of the need for and type of respiratory support) was lower and it could be argued differences in biomarkers may related to severity and may not be specific to SARS-CoV-2 infection'

3. The NC cases were not recruited at the same time as the COVID cases but after (Line 355). Sampling prioritised positive cases first then PUIs after. Study from South Africa has shown that in-hospital mortality was highest during the peaks of the waves when hospital capacity was overwhelmed (Jassat, Lancet Global Health). Not recruiting the comparison group at the same time has potential to bias the analysis, as there could be differences in patient severity and quality of care for those who present during the surge and those who present after.

The purpose was not to compare mortality between NC and COVID-19 cases as this would clearly be confounded. The principal reason to include the NC was to evaluate the potential specificity of white cell counts and inflammatory markers in a contemporaneous group of patients. This reasoning has been added to the methods.

4. The study concurs with other studies that show HIV is not associated with worse outcomes. However this is a small sample single centre study, similar to other studies quoted in the manuscript. A more extensive reading of the literature especially more recent literature suggests an association for HIV and TB with Covid mortality. See WHO report on Covid and HIV, and latest ISARIC Covid hospitalisation analysis.

These references have been added in response to the reviewer's point above.

5. The issue of steroid treatment has been mentioned as a potential confounder, but this would seriously limit the conclusions drawn around inflammatory markers.

We feel we did acknowledge this in the discussion as follows, 'A potential confounder of results on peripheral blood markers is the greater use of corticosteroid therapy at the time of sampling in COVID-19 patients. The results of the Recovery trial were announced shortly after we began recruitment to our study and were accompanied by an overnight change in clinical guidance to prescribe such therapy to those severely ill⁴⁵. Notwithstanding this potential blunting of values, our results again reflect the marked severe systemic effects of acute SARS-CoV-2 infection that are very widely documented'.

The quality of the data

The manuscript employed appropriate methods and analysis, presented findings in appropriate detail, and interpreted findings carefully. My concerns are regarding the small sample size, the description of the study population, particularly controls, differences between covid and NC cases, and potential bias around not recruiting COVID and NC cases at the same time.

We have tried to answer these concerns in our responses above.

The level of support for the conclusions

The main study claims are justified by the data and analytic methods used. However the small sample size and bias may be problematic. The interpretation of no association of HIV with mortality may not be supported due to small sample size and conflicting evidence from large population and hospital cohorts.

We have tried to answer these legitimate concerns in our responses above.

The potential significance of the results

The findings do contribute to broader understanding of HIV, TB co-infection with SARS-CoV-2, and their clinical presentation and outcome. The evidence and arguments presented support advancement of COVID-19 understanding especially for high burden HIV and TB countries, and for Africa where burden of HIV and TB is high and diagnosis of SARS-CoV-2 is likely under-ascertained. It also provides caution in these settings for maintaining clinical vigilance for undetected TB.

We thank the reviewer for recognising important points of emphasis that we tried to communicate.

Your expertise

The interpretation of immune response data is outside the scope of my expertise, and I was unable to assess fully.

Suggested improvements

I have made the following minor suggestions that could help strengthen the work and make it suitable for publication in the journal.

- Suggest revise the title to make it more concise and explanatory

The title is only 88 characters and contains the keywords co-morbid, HIV-1, tuberculosis and SARS-CoV-2 whose inter-relationships we wish to highlight. The title also contains the word Africa from which continent relatively few analyses of this nature are available.

- Use HIV or PLHIV rather than “HIV-1”

We prefer the use of HIV-1 to differentiate it from HIV-2 but if the editor prefers the use of HIV or PLHIV alone we can make this alteration easily.

- Use sub headings in abstract

As mentioned above, we have created subheadings Rationale, Methods, Results and Conclusions.

- Abstract line 45 states this is an observational case cohort; the abstract does not adequately describe the methods and does not state that the patients are sampled from hospitalised patients

We have added this distinction to the abstract ‘by means of an observational case cohort of SARS-CoV-2 patients **sampled from hospitalised patients** stratified by HIV-1 and tuberculosis status’

- Line 87: proportion who died is 2.8% not 28%, check this calculation

We agree, thank the reviewer for pointing this out, and have corrected it.

Reviewer #4 (Remarks to the Author):

Chest radiographs at enrollment were evaluated by three different scoring systems: Brixia, BSTI and percentage of unaffected lung. Only one of the authors is a radiologist. It would be important to know if this author was involved in the interpretation of the images. It would be also important to know if more authors interpreted the images and the degree of agreement between different readers and how the agreement was achieved.

Chest radiographs were reported blind to clinical status by an experienced radiologist (QS-H)

BRIXIA (ref. # 32) refers to a chest radiography score system that examines severity of SARS-CoV-2 disease with age and sex. A more recent paper by the same author (PMID 33263159) describes that Brixia score correlates strongly with disease severity and outcome

BSTI score (BTSICXR) classifies radiographies as classical, indeterminate, and not covid typical images based on imaging patterns. Reference # 33 refers to the ability of “radiographers” to adequately use and apply the BSTI classification system when reporting COVID-19 chest radiographs. This reference may not be adequate for the purpose of this manuscript

We agree and have changed the reference to the original BSTI statement (i.e. Rodrigues JCL, Hare SS, Edey A, Devaraj A, Jacob J, Johnstone A, McStay R, Nair A, Robinson G. An update on COVID-19 for the radiologist - A British society of Thoracic Imaging statement. *Clin Radiol* 2020; 75: 323-325). We are aware there have been several updates and prospective evaluations but it is this statement that guided our evaluation in this study.

“Percentage of unaffected lung” is not explained in the methods. A reference is not given

Percentage unaffected lung is a calculation of our own design. As explained in the methods Posteroanterior chest radiographs were assessed for the total percentage of the lung fields unaffected by any visible pathology. Thus, in the COVID-19 group this score quantified the percentage of normal lung that was not visibly affected by known features of COVID-19 pneumonia on the radiograph. In those with tuberculosis, or in NC with other respiratory infections this score similarly quantified the percentage of normal lung, not visibly affected by the relevant pathology on the radiograph. Those with a normal chest radiograph would thus score 100%.

If the reviewer feels the use of this unaffected lung score adds little to the manuscript that is not provided by BSTI or Brixia we would be happy to remove this section altogether.

Figure 1, A1. It is not clear why CT images are relevant in this manuscript

The images are provided to illustrate case details of overlap between tuberculosis and COVID-19

“Bibasal wedge shape opacities” are not always indicative of pulmonary emboli. This is a CT angiogram which should be able to determine pulmonary artery filling defects, which are more indicative of pulmonary emboli

'The 'bibasal wedge-shaped opacities' describes the appearance on the chest radiograph, in which case the differential diagnosis included pulmonary emboli with infarcts in addition to COVID-19 pneumonia. The CTPA was requested because of persistent hypoxia, and CT confirmed the peripheral 'bibasal wedge-shaped opacities' and presumably confirmed pulmonary emboli. If the emboli involved the areas of opacification it would have been presumed to represent pulmonary infarcts rather than just COVID-19 pneumonia.

Fig B1 A. "diffuse bilateral shadowing:" in not an appropriate radiology term.

Perhaps a more appropriate description would be 'diffuse bilateral pulmonary opacification' with ground glass and consolidation and the manuscript has been changed to reflect this.

Fig B2. This lesion described is nor clearly a cavitary lesion and the upper arrow does not clearly show enlarged subcarinal or precranal lymph nodes

The anterior arrow is pointing to a pulmonary artery so we acknowledge it does NOT show enlarged lymph nodes and apologise for this mistake. The remaining arrow does however pointing to an area of cavitation in the apical segment of the Right Lower Lobe containing air and some fluid.

Reviewers' Comments:

Reviewer #1:

Remarks to the Author:

The authors have adequately addressed my queries

Reviewer #2:

Remarks to the Author:

du Bruyn et al. adequately addressed my critiques but need to be clearer about the limitations of the cohort. Some of the referencing can also be updated:

1) While it is possible that people living with HIV or HIV/TB have milder lung Covid-19 disease by Brixia and WHO ordinal scale, other explanations are possible and should be highlighted. Since the cohort consisted of patients which were admitted to hospital, as opposed to a randomized community sample, an alternative explanation may be that people with HIV or HIV/TB may be better linked to care because they engage with the health care system when receiving ART. Therefore, they may be referred an earlier/milder stage of Covid-19 disease rather than coming in at the point of severe respiratory distress.

2) With the neutralization data, the status of HIV suppression in people living with HIV needs to be taken into account, or if the information is not available, addressed as a limitation. There are a few studies showing that those with HIV viremia make lower neutralizing responses to SARS-CoV-2 exposure (PMID: 34893824, PMID: 35975942). This is consistent with what the authors show for their participants who have not yet initiated ART and are presumably HIV viremic. They should also consider citing a study with similar results on HIV status and SARS-CoV-2 T cell response (PMID: 34611163).

Minor points:

- 1) Figure 1 legend seems to be missing a description of panel E
- 2) Figure 1C, first row (titles) can be confusing as the + sign from the HIV is closer to TB. It can be misinterpreted as (HIV -) or (+/TB+) instead of TB+ (any HIV status)
- 3) Abstract lines 60-61: "...clinical features can be dominated by either SARS-CoV-2 or tuberculosis...". Do the clinical features described indicate TB or "non-Covid like"?

Reviewer #3:

Remarks to the Author:

The authors have adequately addressed the reviewers comments.

Reviewer #5:

Remarks to the Author:

The authors have appropriately addressed the raised points

REVIEWERS' COMMENTS

Reviewer #1 (Remarks to the Author):

The authors have adequately addressed my queries

Reviewer #2 (Remarks to the Author):

du Bruyn et al. adequately addressed my critiques but need to be clearer about the limitations of the cohort. Some of the referencing can also be updated:

1) While it is possible that people living with HIV or HIV/TB have milder lung Covid-19 disease by Brixia and WHO ordinal scale, other explanations are possible and should be highlighted. Since the cohort consisted of patients which were admitted to hospital, as opposed to a randomized community sample, an alternative explanation may be that people with HIV or HIV/TB may be better linked to care because they engage with the health care system when receiving ART. Therefore, they may be referred an earlier/milder stage of Covid-19 disease rather than coming in at the point of severe respiratory distress.

Only those deemed sufficiently unwell to require hospitalisation were included in the study. We acknowledge this as a theoretical possibility. However several reviews have documented decreases in HIV testing, positive HIV tests, and initiation of antiretroviral therapy (ART) during the first lockdown during which this study was conducted (e.g. Jardim Int J Environ Res Public Health 2022 Sep 21;19(19):11899 PMID 36231201; Roy Int J Environ Res Public Health 2022 Aug 8;19(15):9766 PMID 35955119). We have already acknowledged the study design is subject to seen and unseen confounders.

2) With the neutralization data, the status of HIV suppression in people living with HIV needs to be taken into account, or if the information is not available, addressed as a limitation. There are a few studies showing that those with HIV viremia make lower neutralizing responses to SARS-CoV-2 exposure (PMID: 34893824, PMID: 35975942). This is consistent with what the authors show for their participants who have not yet initiated ART and are presumably HIV viremic. They should also consider citing a study with similar results on HIV status and SARS-CoV-2 T cell response (PMID: 34611163).

Only a minority of the patients were not receiving ART which we highlighted in Figure 4C. This precludes meaningful statistical analysis and this limitation is already acknowledged in the discussion 'This feature also precludes meaningful and well-powered comparisons according to ART status'. We thank the reviewer for drawing attention to two additional references published subsequent to our submission of the first revision. Khan et al. (PMID: 34893824) present an important analysis of responses to vaccination whereas the neutralisation data we present is in relation to natural infection. Hwa et al. (PMID: 35975942) interestingly document decreased ability to neutralise the Beta variant of SARS-CoV-2 amongst 11 HIV-1 co-infected participants not prescribed ART. Recruitment to our study ended on August 28th 2020 which was before this variant was detected in the Western Cape. Alrubayyi et al. (PMID 34611163) specifically state they studied Forty-seven individuals with HIV infection in Europe who were well-controlled on

ART (for >2 years) with an undetectable HIV RNA and who were convalescent following mild COVID-19. Direct comparison with our study population is thus fraught. Notwithstanding and in deference to the reviewer's expertise and diligence we have added citations to these studies in the discussion.

Minor points:

1) Figure 1 legend seems to be missing a description of panel E

Checked and corrected

2) Figure 1C, first row (titles) can be confusing as the + sign from the HIV is closer to TB. It can be misinterpreted as (HIV -) or (+/TB+) instead of TB+ (any HIV status)

Figure amended as suggested

3) Abstract lines 60-61: "...clinical features can be dominated by either SARS-CoV-2 or tuberculosis...". Do the clinical features described indicate TB or "non-Covid like"?

'dominated by' changed to 'typical of'

Reviewer #3 (Remarks to the Author):

The authors have adequately addressed the reviewers comments.

Reviewer #5 (Remarks to the Author):

The authors have appropriately addressed the raised points